# DaWin: Training-free Dynamic Weight Interpolation for Robust Adaptation

**Changdae Oh**[1*]**, Yixuan Li**[1]**, Kyungwoo Song**[2†]**, Sangdoo Yun**[3†]**, Dongyoon Han** [3†]

[1]University of Wisconsin–Madison   [2]Yonsei University   [3]NAVER AI Lab

{changdae,sharonli@cs.wisc.edu}    kyungwoo.song@yonsei.ac.kr
{sangdoo.yun,dongyoon.han@navercorp.com}

## Abstract

Adapting a pre-trained foundation model on downstream tasks should ensure robustness against distribution shifts without the need to retrain the whole model. Although existing weight interpolation methods are simple yet effective, we argue their static nature limits downstream performance while achieving efficiency. In this work, we propose **DaWin**, a training-free **d**ynamic **w**eight **in**terpolation method that leverages the entropy of individual models over each unlabeled test sample to assess model expertise, and compute per-sample interpolation coefficients dynamically. Unlike previous works that typically rely on additional training to learn such coefficients, our approach requires no training. Then, we propose a mixture modeling approach that greatly reduces inference overhead raised by dynamic interpolation. We validate DaWin on the large-scale visual recognition benchmarks, spanning 14 tasks across robust fine-tuning – ImageNet and derived five distribution shift benchmarks – and multi-task learning with eight classification tasks. Results demonstrate that DaWin achieves significant performance gain in considered settings, with minimal computational overhead. We further discuss DaWin's analytic behavior to explain its empirical success. Here is our code.

## 1 Introduction

The emergence of foundation models (Bommasani et al., 2021; Radford et al., 2021; Brown et al., 2020) has significantly lowered the barrier to deploying artificial intelligence solutions across a wide range of real-world problems. Leveraging the strong general knowledge acquired through large-scale pre-training, foundation models can be efficiently adapted for numerous tasks. However, recent studies have shown that while fine-tuning improves performance on specific downstream tasks, it may often undermine the model's generalizability and robustness (Wortsman et al., 2022b). For example, a model fine-tuned on ImageNet has better accuracy on in-distribution (ID) data yet may underperform in out-of-distribution (OOD) data such as ImageNet-A (Hendrycks et al., 2021b).

To address this issue, robust fine-tuning methods (Wortsman et al., 2022b) have been recently developed to adapt models to ID while maintaining strong OOD generalization. Some approaches incorporate regularization into the learning objective (Ju et al., 2022; Tian et al., 2023a; Oh et al., 2024), while others focus on preserving the knowledge of the pre-trained model by modifying fine-tuning procedure (Kumar et al., 2022; Lee et al., 2023; Goyal et al., 2023). Notably, *weight interpolation* approaches allow for the integration of knowledge from multiple models via simple interpolation or averaging and have proven effective in both robust fine-tuning (Wortsman et al., 2022b;a; Jang et al., 2024) and multi-task learning (Ilharco et al., 2022; Yadav et al., 2023; Yu et al., 2024) settings.

The weight interpolation approaches are particularly appealing because they are easily applied to any fine-tuned model as a post-hoc plug-in method, delivering competitive performance. Most existing works (Wortsman et al., 2022b;a) focus on creating a single merged model using a static global interpolation coefficient $\lambda$ for *all* test samples: $(1 - \lambda)\theta_0 + \lambda\theta_1$, where $\theta_0$ and $\theta_1$ are the weights of two individual models. While these methods efficiently achieve strong performance, we argue that the optimal coefficients vary across input data samples, leaving notable potential for improving the performance of the interpolation-based approaches. Recent studies on dynamic merging (Cheng et al., 2024; Lu et al., 2024; Tang et al., 2024) explore sample-wise interpolation with $(1 - \lambda(x))\theta_0 +$

---

*Work done during an internship at NAVER AI Lab. † Equal correspondence.

$\lambda(x)\theta_1$, which introduce extra learnable modules to replace the global coefficient with sample-wise coefficient $\lambda(x)$ from a given sample $x$. Yet, these methods require additional training and careful design of the router modules to determine $\lambda(x)$, which brings non-trivial complexity.

In this work, we propose a **d**yna**m**ic **w**eight **in**terpolation framework, **DaWin**, that performs sample-wise interpolation *without any additional training*. To begin with, we conduct a pilot study to investigate the upper-bound performance of dynamic interpolation methods. Specifically, we simulate an oracle sample-wise weight interpolation by leveraging ground truth test labels for sample-wise model expertise estimation via the cross-entropy (X-entropy) loss. Given a test sample, a model yielding a smaller X-entropy incurs larger importance for that sample during weight interpolation, reflecting the expertise of the corresponding model. Here, we observed that fine-grained dynamic interpolation (*e.g.*, at a sample level) indeed significantly outperforms static interpolation.

Our pilot study informatively guides our methodology design by approximating the upper-bound performance in practical scenarios when the ground truth labels of test samples are not available. Specifically, we design an entropy ratio-based score function that can act as a reliable alternative to the X-entropy ratio to robustly determine the sample-wise interpolation coefficients across different samples from diverse domains. The rationale behind using entropy as a surrogate is grounded in the observation that the entropy of a sample's predicted probability distribution strongly correlates with X-entropy. Lastly, to resolve the computation overhead induced by sample-wise interpolation operation during inference time, we further devise a mixture modeling-based (Ma & Leijon, 2011) coefficient clustering method that dramatically reduces the computation. Compared with the most competitive baseline, DaWin improves the performance by 4.5% and 1.8% in terms of the OOD accuracy and multi-task learning average accuracy for CLIP (Radford et al., 2021) image classification, even though DaWin requires far less computational cost during the training or inference.

Our contributions can be summarized in three key points:

i. We present an intuitive numerical analysis of oracle dynamic interpolation methods and show that the X-entropy ratio is a reliable metric to compute per-sample interpolation coefficients.

ii. We propose a practical implementation, DaWin, that approximates the oracle dynamic interpolation by leveraging the prediction entropy ratio of individual models on unlabeled test samples to determine sample-wise interpolation coefficients automatically.

iii. Extensive validation shows that DaWin consistently improves classification accuracy on distribution shift and multi-task learning setups while not remarkably increasing the inference time, and we provide a theoretical analysis to explain the empirical success of DaWin.

## 2 PRELIMINARY

### 2.1 BACKGROUND

**Classification under distribution shift.** We consider a classification problem over a domain $\mathcal{X}$ with input $x$ and a label space $\mathcal{Y} = \{1, ..., C\}$, where the goal is to approximate the true labeling function $h : \mathcal{X} \rightarrow \mathcal{Y}$ with a parametric model $f : \mathcal{X} \rightarrow \Delta^{C-1}$. This model maps $x$ to $C - 1$ dimensional simplex, which aims to minimize error $l : \mathcal{Y} \times \mathcal{Y} \rightarrow \mathbb{R}$ on inputs drawn from any potential target distribution $P_T$. In the robust fine-tuning scenario (Wortsman et al., 2022b; Kumar et al., 2022), the target distribution $P_T$, where a pre-trained model $f$ being adapted to, is typically assumed to be covariate-shifted version (OOD) of the ID source distribution $P_S$. Both distributions share the same class label space $\mathcal{Y}$, and have the same conditional distribution over target labels, i.e., $P_S(y|x) = P_T(y|x)$, but have different marginal distributions over input $P_S(x) \neq P_T(x)$.

**Model merging.** Let $f(\cdot; \theta_0)$ and $f(\cdot; \theta_1)$ denote models that are individually trained on the same or different datasets but have identical architecture. For example, $f(\cdot; \theta_0)$ could represent a pre-trained model such as CLIP (Radford et al., 2021), while $f(\cdot; \theta_1)$ is the fine-tuned counterpart on a particular downstream task. Model merging approach (Wortsman et al., 2022b;a) constructs a merged model $f(\cdot; \theta_\lambda)$, which achieves a better trade-off between ID and OOD performance than the individual models by interpolating in the weight space $\theta_\lambda = (1 - \lambda)\theta_0 + \lambda\theta_1$. We will use the terms interpolation and merging interchangeably. Throughout the paper, we use the term *static interpolation* to denote methods that induce a single merged model corresponding to a single interpolation coefficient applied for all test samples. In contrast, *dynamic interpolation* refers to methods that yield multiple merged models, with interpolation coefficients depending on the sample $x$ or domain $\mathcal{X}$.

Table 1: **Pilot experiments for upper-bound analysis**. We evaluate zero-shot (ZS), fine-tuned (FT), and several interpolation methods with CLIP ViT-B/32 on ImageNet (IN) and its five distribution shifts: ImageNet (IN)-V2/R/A/S and ObjectNet (ObjNet). Given test domain $\mathcal{X}$, domain-wise coefficients $\lambda^*(\mathcal{X})$ are found by grid search over each test set, and the sample-wise coefficients $\lambda^*(x)$ for each test sample $x$ are determined by the X-entropy loss ratio of individual models. † denotes *oracles* that utilize ground truth test labels.

| Method | Model Weight | Acc. Under Distribution Shifts | | | | | | |
|---|---|---|---|---|---|---|---|---|
| | | IN | IN-V2 | IN-R | IN-A | IN-S | ObjNet | Avg |
| ZS (Radford et al., 2021) | $\theta_0$ | 63.4 | 55.9 | 69.3 | 31.4 | 42.3 | 43.5 | 48.5 |
| FT (Wortsman et al., 2022b) | $\theta_1$ | 78.4 | 67.2 | 59.3 | 24.7 | 42.2 | 42.0 | 47.9 |
| WiSE-FT (Wortsman et al., 2022b) | $(1-\lambda)\theta_0 + \lambda\theta_1$ | 79.1 | 68.4 | 65.4 | 29.4 | 46.0 | 45.9 | 51.0 |
| Dynamic Interpolation† (domain) | $(1-\lambda^*(\mathcal{X}))\theta_0 + \lambda^*(\mathcal{X})\theta_1$ | 79.1 | 68.5 | 72.9 | 36.3 | 48.5 | 48.9 | 55.0 |
| Dynamic Interpolation† (sample) | $(1-\lambda^*(x))\theta_0 + \lambda^*(x)\theta_1$ | **83.4** | **74.4** | **77.9** | **42.9** | **53.4** | **54.6** | **60.6** |

Figure 1: **Distribution of interpolation coefficients**. Histograms of sample-wise interpolation coefficients induced by the X-entropy ratio between two models (ZS and FT). Oracle coefficient estimates vary significantly within a domain (IN) and across domains (IN v.s. IN-V2/R/A/S/ObjNet) regarding symmetry and skewness.

## 2.2 PILOT STUDY

**Our goal.** We begin by conducting a pilot study to understand the benefits of dynamic interpolation, which adapts interpolation coefficients at a finer granularity (such as sample level, $\lambda(x)$), as opposed to using global coefficient $\lambda$ for all samples (Wortsman et al., 2022b;a). To explore the upper limits of these methods, we experiment with ground truth labels as an *oracle*[1] to estimate upper-bound performance and understand the maximum potential of model interpolation methods. This study will further guide our method design in Sec. 3 by approximating the oracle performance.

**Setup and hypothesis.** We assess the top-1 classification accuracy of several approaches on ImageNet-1K (IN) (Russakovsky et al., 2015) and distribution-shifted benchmarks IN-V2/R/A/S/ObjNet (Recht et al., 2019; Hendrycks et al., 2021a;b; Wang et al., 2019; Barbu et al., 2019) by fine-tuning the CLIP ViT-B/32[2] (Radford et al., 2021) on IN. Besides the individual models (zero-shot; ZS and fine-tuned; FT), we include a representative static interpolation method, WiSE-FT (Wortsman et al., 2022b), which determines the interpolation coefficient $\lambda$ via grid search on the ID validation set. Then, we implement two oracle interpolation methods: Dynamic Interpolation† per domain and per sample. For the domain-wise dynamic interpolation, oracle coefficients $\lambda^*(\mathcal{X})$ are determined by grid search over all test samples within each domain $\mathcal{X}$, such as art or sketch. In contrast, for sample-wise oracle interpolation coefficients $\lambda^*(x)$, we use negative X-entropy to measure models' expertise on a specific input $x$, which results in $\lambda^*(x) = \exp(-l(f(x;\theta_1), y))/(\exp(-l(f(x;\theta_0), y)) + \exp(-l(f(x;\theta_1), y)))$. Note that we can regard the model $f(x;\theta)$ as an estimator of a conditional probability density function, e.g., $p_\theta(y|x)$. Then, given the equality between the X-entropy over the one-hot label and negative log-likelihood, $\lambda^*(x)$ can be interpreted as a posterior of Bernoulli distribution in the *noise-contrastive estimation* (Gutmann & Hyvärinen, 2010), i.e., $\lambda^*(x) = p_{\theta_1}(y|x)/(p_{\theta_0}(y|x) + p_{\theta_1}(y|x))$, which models the probability whether a data $(x, y)$ comes from the distribution $p_{\theta_1}(y|x)$. See Appendix A.1 for more.

In this experiment, we hypothesize that (1) Fine-grain interpolation, which adapts coefficients at a finer level (such as the sample level), can lead to substantial improvements compared to static interpolation, such as WiSE-FT. (2) Per-sample interpolation coefficients can effectively estimated with X-entropy-based model expertise measurement.

---

[1] We use the term *oracle* to denote a scenario where we utilize the ground truth test label, and it should not be confused with *optimal*. Optimality of $\lambda^*(x)$ regarding weight interpolation is further discussed in Tab C.

[2] We employed the pre-trained model available at: https://github.com/openai/CLIP.

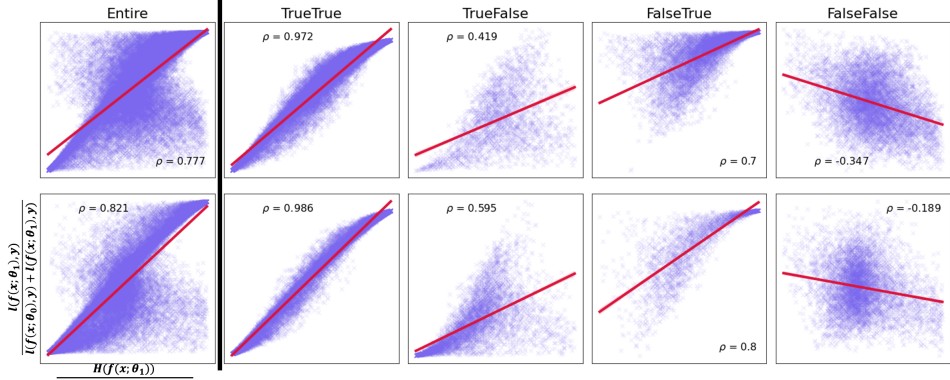

Figure 2: **Correlation between entropy ratio and X-entropy ratio**. We split each test set into four subsets (`TrueTrue`, `TrueFalse`, `FalseTrue`, `FalseFalse`) based on the correctness of two models' predictions (ZS and FT). On each split and the entire data, we show scatter plots of the entropy ratio (x-axis) and x-entropy ratio (y-axis) computed by two models with corresponding Pearson correlations. The top and bottom rows denote results from ImageNet and ImageNet-R. Results imply that the entropy ratio strongly correlates with the X-entropy ratio overall, except `FalseFalse` case, wherein we could not expect satisfactory performance for individual models and interpolation methods as an inherent limitation given fixed individual models.

**Results and interpretations.** We verify our hypothesis in Table 1. Here, the domain-wise and sample-wise dynamic interpolation methods perform far better than individual models or static interpolation. Moreover, compared with the domain-wise interpolation method, sample-wise interpolations show much higher performance in all cases. This supports our hypotheses that fine-grain interpolation leads to better downstream accuracy, and the negative X-entropy can be used as a reliable estimator of per-sample model expertise to induce the interpolation coefficients.

In Figure 1, the interpolation coefficients $\lambda^*(x)$, computed using the X-entropy ratio (corresponding to the last row of Table 1), vary significantly within and across domains. On the one hand, the estimated coefficients are widely distributed from zero to one on all evaluation datasets. On the other hand, the distributions of coefficients also vary depending on the domain, e.g., IN and IN-A exhibit right-skewed and left-skewed distributions, respectively. This confirms that optimal coefficient estimation varies by sample, motivating the exploration of the sample-wise dynamic interpolation method. In summary, we conclude that ***determining proper interpolation coefficients on a sample-wise basis dramatically elevates the achievable performance of model merging-based approaches***.

## 3 METHOD

### 3.1 REVISITING ENTROPY AS A MEASURE OF MODEL EXPERTISE

While the pilot study in the previous section provides encouraging results, those oracle methods cannot work in practice. This is because we do not have access to the ground truth labels for incoming test samples. This leads us to the question: *how can we reliably estimate the interpolation coefficient solely based on the test input x?* There is extensive literature which adopts **entropy as a proxy of X-entropy** (Grandvalet & Bengio, 2004; Chapelle & Zien, 2005; Shu et al., 2018; Wang et al., 2021; Prabhu et al., 2021). The rationale behind using entropy as a surrogate is grounded in the observation that the entropy of a sample's prediction distribution strongly correlates with X-entropy, even under distribution shifts (Wang et al., 2021) those models have not explicitly trained on. This work presents the first attempt to leverage the sample-wise entropy to measure each model's expertise to determine the interpolation coefficients given each test sample. It is worth noting that this approach differs from AdaMerging (Yang et al., 2024b), which seeks to learn a global coefficient inducing a single merged model to minimize the expected entropy over the entire test set.

In this section, we will show that entropy is well-correlated with X-entropy and can thus be used to estimate model expertise. Specifically, we divide the test set into four splits, {`TrueTrue`, `TrueFalse`, `FalseTrue`, `FalseFalse`}, based on the correctness of predictions made by two models (zero-shot and fine-tuned CLIP). For example, `TrueTrue` represents the set $\{(x_i, y_i)|f(x_i; \theta_0) = y_i \text{ and } f(x_i; \theta_1) = y_i\}$. We then compute the Pearson correlation coeffi-

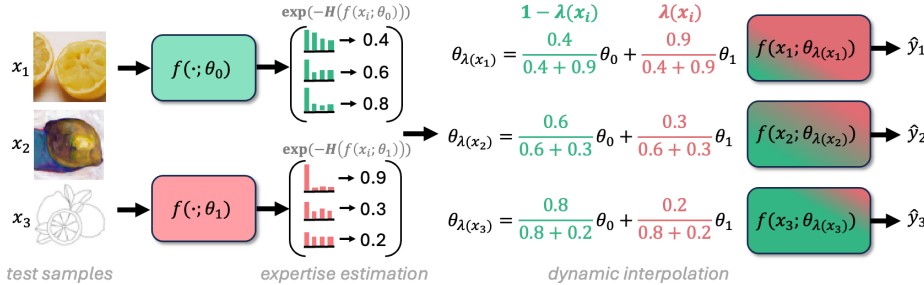

Figure 3: **Framework overview**. DaWin estimates per sample model expertise (exponentiated negative entropy) and then produces coefficients based on the relative expertise of models for dynamic interpolation.

cients between entropy and X-entropy within each split. Figure 2 shows scatter plots with fitted regression lines of the entropy and X-entropy ratios between two models on IN (top) and IN-R (bottom). Specifically, we plot $\frac{H(f(x;\theta_0))}{H(f(x;\theta_0))+H(f(x;\theta_1))}$ and $\frac{l(f(x;\theta_0),y)}{l(f(x;\theta_0),y)+l(f(x;\theta_1),y)}$ for entire test samples and compute Pearson correlation coefficient between them, where $H$ denotes the entropy.

Here, aside from the FalseFalse case where both models failed to produce reliable predictions[3], we observe strong correlations (Schober et al., 2018) across the remaining splits and the entire dataset. This implies that entropy is a reasonable proxy to the X-entropy for estimating per-sample model expertise and, ultimately, computing sample-wise interpolation coefficients.

## 3.2 DYNAMIC INTERPOLATION VIA ENTROPY RATIO

After confirming a strong correlation between the entropy and X-entropy, we propose a new dynamic weight interpolation, DaWin, by defining sample-wise interpolation coefficients $\lambda(x)$ as below:

$$\lambda(x) = \frac{\exp(-H(f(x;\theta_1)))}{\exp(-H(f(x;\theta_0))) + \exp(-H(f(x;\theta_1)))}. \tag{1}$$

Here, $H$ represents the entropy over the output probability distribution of model $f(\cdot;\theta)$ given input $x$. The value of $\lambda(x)$ approaches 1 when model $f(\cdot;\theta_1)$ exhibits lower entropy (greater expertise), whereas it approaches 0 if the entropy of model $f(\cdot;\theta_1)$ becomes higher. Figure 3 illustrates the overall framework. Intuitively, this approach is analogous to classifier selection methods (Giacinto & Roli, 2001; Ko et al., 2008), which aims to dynamically select suitable classifier(s) given a test sample. However, DaWin differs in terms of its motivation, expertise metric, and goal as we pursue finding the interpolation coefficients that may induce better performance compared with the best model selection method (See Table 5). We further discuss the connection between DaWin and the selection-based method in Sec. 6. Note that our sample-wise expertise estimations **do not require any hyperparameters** for computing interpolation coefficients. This eliminates the need for intensive hyperparameter tuning often required in prior works (Wortsman et al., 2022b; Ilharco et al., 2023) to achieve the best result. Meanwhile, if we assume access to all test samples simultaneously, we can refine some unstable coefficient estimations by introducing per-domain expertise offset. We use this offset adjustment by default (Sec. A.1). Besides, to bypass using overconfident output (Guo et al., 2017) during expertise estimation, we conduct the temperature scaling on ID validation set.

**Sample-wise entropy valley hypothesis**. One of the most popular hypotheses that explain the success of weight interpolations is *linear mode connectivity* (LMC; Frankle et al. (2020)), which ensures that interpolated models are also laid on good solution space as well as low-loss converged individual models. Although entropy is a concave function for the probability density, we believe that a similar statement can be claimed, which we refer to *sample-wise entropy valley* as follows:

i) There exists $\lambda$ such that $\mathbb{E}_x[H(f(x;\lambda\theta_0 + (1-\lambda)\theta_1)] \leq \min\{\mathbb{E}_x[H(f(x;\theta_0))], \mathbb{E}_x[H(f(x;\theta_1))]\}$.

ii) Given $\lambda$ obeyed to (i) and evidence of sample-wise linear feature connectivity (Zhou et al., 2023), there also exists $\lambda(x)$ such that $\mathbb{E}_x[H(f(x;\lambda(x)\theta_0 + (1-\lambda(x))\theta_1)] \leq \mathbb{E}_x[H(f(x;\lambda\theta_0 + (1-\lambda)\theta_1)]$.

As empirical validation of this hypothesis, Figure 6 and Figure H show that the interpolated model actually induces smaller entropy than individuals (i) on average, and DaWin achieves lower entropy than static interpolation (ii). Given the strong correlation between entropy and X-entropy, this hypothesis advocates the benefit of DaWin than static interpolation methods.

[3]However, even individuals fail, an interpolated model can yield correct predictions (Yong et al., 2024).

### 3.3 Efficient Dynamic Interpolation by Mixture Modeling

Although our primary goal is to achieve better downstream task performance by way of dynamic interpolation, performing interpolation on every sample inevitably increases the inference-time computation, with the cost scaling linearly with the number of model parameters (Lu et al., 2024). To ensure practicality while pursuing state-of-the-art downstream task performance, we devise an efficient dynamic interpolation method by leveraging mixture modeling.

Let $\{x_i\}_{i=1}^N$ denote the $N$ test samples and $\{\lambda(x_i)\}_{i=1}^N$ the corresponding interpolation coefficients computed by DaWin framework. Note that $\lambda(x_i) \in [0, 1]$ for all $i = 1, \ldots, N$, and they can be regarded as sampled observations from some Beta distributions. Then, we accurately model the probability density function of interpolation coefficients via **Beta mixture model**[4] as follows:

$$\Lambda_i \sim \Sigma_{k=1}^K \pi_k \text{Beta}(\lambda(x_i); a_k, b_k),$$

where $\Lambda_i$ indicates random variables of $\lambda(x_i)$ and $\{\pi_k\}_{k=1}^K$ are the prior probabilities for each Beta distribution $\text{Beta}(\cdot; a_1, b_1), ..., \text{Beta}(\cdot; a_K, b_K)$. We initialize the parameters $(a_k, b_k)_{k=1}^K$ randomly for each Beta distribution and estimate the initial membership of each sample via $K$-means clustering. Then, the expectation-maximization (EM) algorithm is adopted to iteratively refine the parameters of the Beta mixture and the membership inference quality. This process only takes around 30 seconds for 50K samples. Finally, given a test sample $x_i$, we infer the membership of that sample via maximum likelihood $k^* = \arg\max_k \text{Beta}(\lambda(x_i); a_k, b_k)$, and use the mean of the corresponding Beta distribution $\frac{a_{k^*}}{a_{k^*} + b_{k^*}}$. This approach significantly reduces the computational burden of sample-wise dynamic interpolation from $N$ (number of test samples) to $K$ (number of pre-defined clusters), where $K \ll N$. We outline the detailed procedures of DaWin in Algorithm 1 (Appendix A.1).

## 4 Experiments

### 4.1 Setup

**Tasks and datasets.** We validate DaWin on two scenarios: (1) **robust fine-tuning** (Wortsman et al., 2022b) and (2) **multi-task learning** (Ilharco et al., 2022) with focusing on the top-1 classification accuracy (Acc). Following robust fine-tuning literature (Wortsman et al., 2022b; Kumar et al., 2022), we use ImageNet-1K and its five OOD variants, ImageNet-V2, ImageNet-R, ImageNet-A, ImageNet-Sketch, and ObjectNet, to evaluate robustness under distribution shifts. For multi-task learning, we follow the standard evaluation protocol (Ilharco et al., 2022; Yang et al., 2024b) using eight benchmark datasets: SUN397, Cars, RESISC45, EuroSAT, SVHN, GTSRB, MNIST, and DTD. Please see the Appendix A.2 for the omitted references and description for each dataset.

**Models and baselines.** For robust fine-tuning, we adopt CLIP ViT-{B/32, B/16, L/14} (Radford et al., 2021) as zero-shot (ZS) models to ensure a fair comparison with (Wortsman et al., 2022b;a; Ilharco et al., 2022; Yang et al., 2024b), and fine-tuned (FT) checkpoints for each backbone from Jang et al. (2024) (see details in Appendix A.2). For the weight interpolation baseline methods, we include WiSE-FT (Wortsman et al., 2022b), Model Soup (Wortsman et al., 2022a), and Model Stock (Jang et al., 2024), along with the traditional output ensemble method. In addition, we consider several state-of-the-art robust fine-tuning methods such as LP-FT (Kumar et al., 2022), CAR-FT (Mao et al., 2024), FLYP (Goyal et al., 2023), Lipsum-FT (Nam et al., 2024), and CaRot (Oh et al., 2024). For multi-task learning, we use CLIP ViT-B/32 as our backbone and consider the model merging baselines as follows: weight averaging, Fisher Merging (Matena & Raffel, 2022), RegMean (Jin et al., 2023), Task Arithmetic (Ilharco et al., 2023), Ties-Merging (Yadav et al., 2023), AdaMerging (Yang et al., 2024b), and Pareto Merging (Chen & Kwok, 2024). For a fair comparison, we do not experiment with other training-intensive dynamic methods that introduce lots of additional parameters (Tang et al., 2024), the post-hoc methods (Yang et al., 2024a) and test-time prompt tuning (Shu et al., 2022), which are orthogonal to baseline and our methods. For DaWin, we set $K$ to 3, 5, 2 for ViT-{B/32, B/16, L/14} in the robust fine-tuning and $K = 1$ in the multi-task setups.

### 4.2 Results on Robust Fine-tuning

**Better performance trade-off between ID and OOD.** In Figure 4, we present the accuracy for ID ($x$-axis) and OOD average ($y$-axis) across three backbone models, along with baseline performance.

---

[4]We advocate Beta mixture more than a non-parametric method, e.g., $K$-means, for better performance due to the distributional structure. For cases of more than two models, it can be easily extended to Dirichlet mixture.

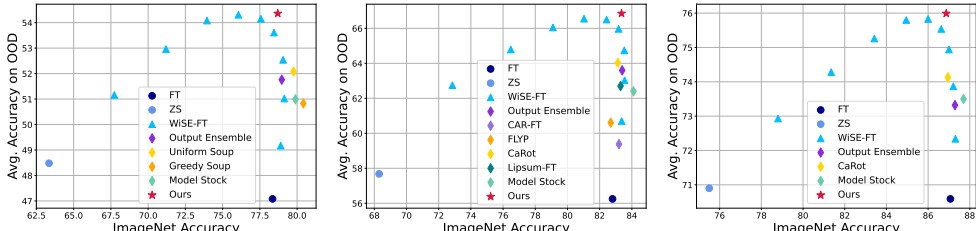

Figure 4: **ID and OOD trade-off analysis**. Trade-offs in terms of ID and OOD classification accuracy on ImageNet distribution shift benchmarks with CLIP ViT-{B/32, B/16, L/14} from left to right. The hyperparameter grid of WiSE-FT is set to $\{0.1, 0.2, ..., 0.9\}$. `DaWin` achieves performance beyond the Pareto-optimal points given two models, ZS and FT (See Figure A in Appendix A.3 for more results).

Achieving superior ID/OOD trade-offs beyond or even comparable to the WiSE-FT hyperparameter sweep is challenging. Nevertheless, `DaWin` provides a remarkably better performance trade-off compared to baseline methods. It is also worth noting that `DaWin` does not require any hyperparameter tuning. Instead, it dynamically generates interpolation coefficients solely based on the trained models' output. This distinguishes `DaWin` from other hyperparameter-intensive fine-tuning such as CaRot (Oh et al., 2024) and Lipsum-FT (Nam et al., 2024) or existing interpolation methods (Wortsman et al., 2022b). Table 2 provides detailed train/inference costs and ID/OOD accuracies. Compared with Model Soups (Uniform and Greedy Soups) and Model Stock, which require more than two models to ensure diversity among interpolation candidate models (Wortsman et al., 2022a) or periodic interpolations during training (Jang et al., 2024), `DaWin` operates with only a single fine-tuned model while achieving significantly better performance trade-off. These trends hold across other backbones (see Tables 3 and 4), verifying `DaWin`'s generality in terms of modeling scale. Although `DaWin` compromises the inference-time efficiency for accuracy, given that existing works typically require hyperparameter tuning in practice (Wortsman et al., 2022b), relative computation overhead is minor given that we set $(K + M) \lesssim H$ in all our experiments (See Figure 8).

Table 2: Accuracy on ImageNet (ID) and its OOD for CLIP ViT-B/32. Cost (T, I) denote the number of training runs required to build the final model and the number of evaluations during inference, respectively. Given $M$ models, $N$ denotes the number of test samples, and $H$ and $K$ denote the size of the hyperparameter grid and the number of mixture components, respectively. Note that $(K + M) \lesssim H$ in all our experiments. Coefficients for Output ensemble and WiSE-FT were picked by grid search on ID validations set.

| Method | Cost (T) | Cost (I) | ImageNet Acc (ID) | Avg. Acc on OOD |
|---|---|---|---|---|
| ZS | - | $\mathcal{O}(1)$ | 63.35 | 48.48 |
| FT | 1 | $\mathcal{O}(1)$ | 78.35 | 47.08 |
| Output ensemble | 1 | $\mathcal{O}(M)$ | 78.97 | 51.76 |
| WiSE-FT (Wortsman et al., 2022b) | 1 | $\mathcal{O}(H)$ | 79.14 | 51.02 |
| Uniform Soup (Wortsman et al., 2022a) | 48 | $\mathcal{O}(1)$ | 79.76 | 52.08 |
| Greedy Soup (Wortsman et al., 2022a) | 48 | $\mathcal{O}(1)$ | **80.42** | 50.83 |
| Model Stock (Jang et al., 2024) | $2+\alpha$ | $\mathcal{O}(1)$ | 79.89 | 50.99 |
| **DaWin w/o mixture modeling** | 1 | $\mathcal{O}(N + M)$ | 78.71 | **54.41** |
| **DaWin** | 1 | $\mathcal{O}(K + M)$ | 78.70 | 54.36 |

**Ablation study.** We explore alternative metrics beyond entropy for estimating sample-wise model expertise, as shown in Figure 5. Specifically, we consider four different pseudo label (PL) approaches (Lee et al., 2013) to replace the entropy, $\{\hat{y}, \tilde{y}\} \times \{\text{soft}, \text{hard}\}$, where $\hat{y} = \frac{1}{2}f(x; \theta_0) + \frac{1}{2}f(x; \theta_1)$ and $\tilde{y} = f(x; \frac{1}{2}\theta_0 + \frac{1}{2}\theta_1)$. The settings $\{\text{soft}, \text{hard}\}$ indicate whether the `argmax` operation is applied to PL or not. We observe that some PLs slightly outperform entropy in terms of ID accuracy but bring no gains in OOD accuracy. To maintain simplicity in line with Occam's razor (Good, 1977), we adopt entropy as the expertise metric on unlabeled test samples.

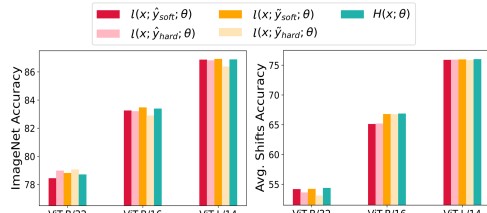

Figure 5: **Effectiveness of our expertise metric**. We evaluate five different expertise metrics on ID (left) and OOD (right). Entropy ($H$) behaves like loss ($l$) with the pseudo labels, where its performance nearly matches or surpasses the best-performing $l$ though its simplicity.

Table 3: Accuracy on ImageNet (ID) and distribution shifts (OOD) for CLIP ViT-B/16.

| Method | ID | OOD Avg. |
|---|---|---|
| ZS | 68.3 | 57.7 |
| FT | 82.8 | 56.3 |
| LP-FT (Kumar et al., 2022) | 82.5 | 61.3 |
| CAR-FT (Mao et al., 2024) | 83.2 | 59.4 |
| FLYP (Goyal et al., 2023) | 82.7 | 60.6 |
| Lipsum-FT (Nam et al., 2024) | 83.3 | 62.7 |
| CaRot (Oh et al., 2024) | 83.1 | 64.0 |
| Output ensemble | 83.4 | 63.6 |
| WiSE-FT (Wortsman et al., 2022b) | 83.5 | 64.2 |
| Model Stock (Jang et al., 2024) | **84.1** | 62.4 |
| **DaWin** | 83.4 | **66.9** |

Table 4: Accuracy on ImageNet (ID) and distribution shifts (OOD) for CLIP ViT-L/14.

| Method | ID | OOD Avg. |
|---|---|---|
| ZS | 75.5 | 70.9 |
| FT | 87.0 | 70.6 |
| FLYP | 86.2 | 71.4 |
| CaRot | 87.0 | 74.1 |
| Output ensemble | 87.3 | 73.3 |
| WiSE-FT | 87.3 | 73.2 |
| Model Stock | **87.7** | 73.5 |
| **DaWin** | 86.9 | **76.0** |

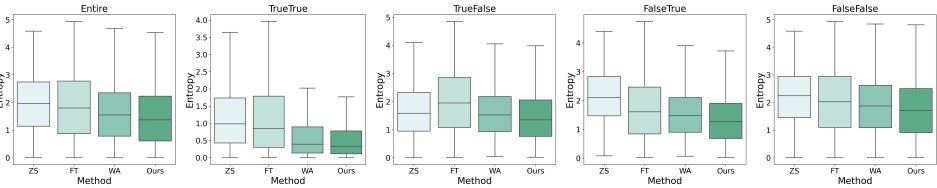

Figure 6: **Entropy comparison on ImageNet-A for CLIP ViT-B/32.** We visualize the average entropy from each method on the entire dataset and four different splits based on the correctness of zero-shot (ZS) and fine-tuned (FT) models. Compared with individual models, weight averaging (WA) induces lower entropy overall, and our `DaWin` achieves the lowest entropy across all splits (See Figure F in Appendix for full results).

**Entropy analysis.** Our `DaWin` method is built on the correlation between entropy and X-entropy. While the analyses from Sec. 2.2 support using entropy as a proxy of X-entropy, it does not answer how the entropy of interpolated final models behaves. To explore this, we analyze the average entropy of interpolated models generated by `DaWin` in Figure 6. We see that the simple weight averaging (WA) produces lower entropy compared to individual models, and our `DaWin` achieves the lower entropy in all cases. This indicates that `DaWin`'s expertise-based interpolation successfully weighs individual experts for accurate prediction and decreases per-sample entropy accordingly. In Sec. 6, we further provide a discussion on the analytic behavior of `DaWin`'s weighting strategy.

**Applications: dynamic selection and dynamic output ensemble.** Given that `DaWin` is motivated by the concept of "*entropy as a measure of model expertise*," we can extend this beyond weight interpolation to two other approaches, dynamic classifier selection (DCS; Giacinto & Roli (2001); Britto et al. (2014)) and dynamic output ensemble (DOE; Alam et al. (2020); Li et al. (2023)).

To be specific, DCS aims to select the most suitable classifier for each test sample based on the *competence* (referred to as *expertise* in our terminology) among multiple classifiers. We provide the results of DCS by adopting entropy as a competence measurement for classifier selection, *i.e.*, selecting a lower entropy model per sample. Besides, we also present a dynamic output ensemble (DOE) by our $\lambda(x)$ directly on the output probability space rather than weight space. In Table 5, we see that entropy-based DCS and DOE bring large performance gains compared with a single fine-tuned model (FT), while the `DaWin` achieves the largest gain. This implies that users can flexibly build their dynamic system using test sample entropy according to their budget and performance criteria.

Table 5: Results of per-sample dynamic classifier selection (DCS), dynamic output ensemble (DOE), and dynamic weight interpolation (`DaWin`) on ImageNet (ID) and under distribution shifts (OOD) for different CLIP ViT backbone models.

| | Model | Method | | | |
|---|---|---|---|---|---|
| | | FT | DCS | DOE | **DaWin** |
| ID | B/32 | 78.35 | 78.59 | **78.71** | **78.71** |
| | B/16 | 82.80 | 82.15 | 83.24 | **83.38** |
| | L/14 | **87.07** | 86.53 | **87.07** | 86.88 |
| OOD | B/32 | 47.08 | 52.87 | 52.71 | **54.41** |
| | B/16 | 56.25 | 64.90 | 64.85 | **66.85** |
| | L/14 | 70.59 | 74.71 | 75.14 | **76.01** |

### 4.3 RESULTS ON MULTI-TASK LEARNING

We now evaluate existing merging approaches and our `DaWin` on multi-task learning benchmarks with CLIP ViT-B/32. Following the standard evaluation protocol (Yang et al., 2024b), we have $M$

Table 6: **Multi-task learning performance.** We use CLIP ViT-B/32 for evaluation across eight benchmark tasks. † denotes using the ground truth domain indicator to select expert models for each domain. We report the layer-wise method for AdaMerging (Yang et al., 2024b) here, which surpasses those of the task-wise method. Note that AdaMerging and Pareto Merging both require a non-trivial amount of training.

| Method | SUN397 | Cars | RESISC45 | EuroSAT | SVHN | GTSRB | MNIST | DTD | Avg. |
|---|---|---|---|---|---|---|---|---|---|
| Pre-trained | 63.2 | 59.6 | 60.2 | 45.2 | 31.6 | 32.6 | 48.3 | 44.4 | 48.1 |
| Jointly fine-tuned | 73.9 | 74.4 | 93.9 | 98.2 | 95.8 | 98.9 | 99.5 | 77.9 | 88.9 |
| Individuals† | 75.3 | 77.7 | 96.1 | 99.8 | 97.5 | 98.7 | 99.7 | 79.4 | 90.5 |
| Weight Average (Ilharco et al., 2022) | 65.3 | 63.3 | 71.4 | 72.6 | 64.2 | 52.8 | 87.5 | 50.1 | 65.9 |
| Fisher Merging (Matena & Raffel, 2022) | 68.6 | 69.2 | 70.7 | 66.4 | 72.9 | 51.1 | 87.9 | 59.9 | 68.3 |
| RegMean (Jin et al., 2023) | 65.3 | 63.5 | 75.6 | 78.6 | 78.1 | 67.4 | 93.7 | 52.0 | 71.8 |
| Task Arithmetic (Ilharco et al., 2023) | 55.3 | 54.9 | 66.7 | 75.9 | 80.2 | 69.7 | 97.3 | 50.1 | 68.8 |
| Ties-Merging (Yadav et al., 2023) | 65.0 | 64.3 | 74.7 | 75.7 | 81.3 | 69.4 | 96.5 | 54.3 | 72.6 |
| AdaMerging (Yang et al., 2024b) | 64.2 | 68.0 | 79.2 | 93.0 | 87.0 | 92.0 | 97.5 | 58.8 | 80.0 |
| AdaMerging++ (Yang et al., 2024b) | 65.8 | 68.4 | 82.0 | 93.6 | 89.6 | 89.0 | 98.3 | 60.2 | 80.9 |
| Pareto Merging (Chen & Kwok, 2024) | **71.4** | **74.9** | 87.0 | 97.1 | 92.0 | 96.8 | 98.2 | 61.1 | 84.8 |
| **DaWin** | 66.2 | 66.7 | **91.3** | **99.2** | **94.7** | **98.1** | **99.5** | **74.6** | **86.3** |

tasks and models individually fine-tuned on each task (where $M = 8$). We do not know where each test sample arises from during evaluation and cannot choose true experts per domain. *While both AdaMerging and* DaWin *use unlabeled testset,* DaWin *produces dynamic merged models given tasks or samples, whereas AdaMerging induces a single merged model for all tasks by default.*

Here, we modify the interpolation formula in Sec. 2.1 into task arithmetic formulation, *i.e.*, given weights of pre-trained model $\theta_0$ and fine-tuned models $\{\theta_j\}_{j=1}^M$, a dynamic interpolation is defined as $\theta_{\lambda(x)} = \theta_0 + \lambda_0 \sum_{j=1}^M \lambda_j(x)\tau_j$ where $\tau_j = \theta_j - \theta_0$, $\lambda_j(x)$, and $\lambda_0$ denote the task vector (Ilharco et al., 2023), weight for $j$-th task vector, and scaling term (set to 0.3 following Ilharco et al. (2023)), respectively. In Table 6, DaWin greatly outperforms advanced weight averaging methods (Jin et al., 2023) and adaptive merging methods (Chen & Kwok, 2024) that require tough training, whereby approaching a ground truth expert selection method, e.g., Individuals†. This verifies the versatility of DaWin, which is beneficial for adapting the model on multiple tasks as well as a single-task setup. Figure 7 shows the average of estimated sample-wise coefficients (y-axis) per dataset (x-axis). DaWin assigns the highest weights to

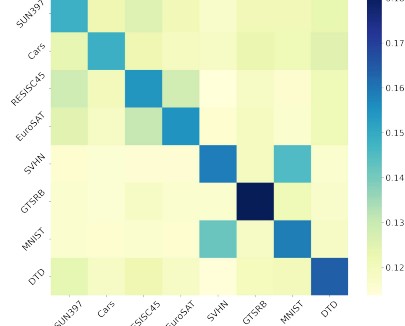

Figure 7: **Merging coefficient visualization.** We visualize DaWin's coefficients across 8 datasets (columns) with 8 fine-tuned models (rows).

true experts (diagonal) per task and leverages relevant experts by reflecting task-wise similarity, e.g., SVHN ⇔ MNIST for digit recognition and EuroSAT ⇔ RESISC45 for scenery classification.

# 5 RELATED WORK

**Robust fine-tuning** aims to adapt a model on a target task while preserving the generalization capability learned during pre-training. A straightforward approach injects regularization into the learning objective. For example, Ju et al. (2022) proposed a regularization motivated by Hessian analysis, Tian et al. (2023a;b) devised a trainable projection method to constrain the parameter space, CAR-FT (Mao et al., 2024) devised the context-awareness regularization, and CaRot (Oh et al., 2024) introduced a regularization based on singular values. Another line of works modifies the training procedure to keep the pre-trained knowledge by decoupling the tuning of a linear head from the entire model (Kumar et al., 2022), employing a data-dependent tunable module (Lee et al., 2023), utilizing bi-level optimization (Choi et al., 2024), or mimicking the pre-training procedure (Goyal et al., 2023). In contrast, weight interpolation approaches emerged as an effective yet efficient solution that conducts simple interpolation of individual model weights (Izmailov et al., 2018; Wortsman et al., 2022a;b; Jang et al., 2024). Unlike existing works inducing a single interpolated model, we propose a dynamic interpolation method that produces per-sample models for better adaptation.

**Model merging** studies mainly focus on integrating multiple models trained on different tasks into a single model to create a versatile, general-purpose multi-task model. After some seminal works

in the era of foundation models (Ilharco et al., 2022; 2023; Jin et al., 2023), numerous advances have been made those aim at reducing conflict between merged parameters (Yadav et al., 2023; Yu et al., 2024; Marczak et al., 2024), merging with weight disentanglement (Wang et al., 2024; Ortiz-Jimenez et al., 2024; Jin et al., 2024), optimizing interpolation coefficients on unlabeled test samples (Yang et al., 2024b; Chen & Kwok, 2024) or learning additional modules generating per-sample/domain coefficients dynamically (Cheng et al., 2024; Lu et al., 2024; Tang et al., 2024; Yang et al., 2024a). While those methods provide huge performance gains compared with static methods such as (Wortsman et al., 2022a), they all bring extra learnable modules and training. We focus on methods that do not induce extra complex training and devise a training-free dynamic merging method, `DaWin`, which seeks a good trade-off between efficiency and downstream performance.

## 6 DISCUSSION

**Accuracy and runtime trade-off.** While we focus on improving accuracy through model merging approaches, it is crucial to ensure the extra computation demands of `DaWin` during inference are manageable. Fig. 8 presents trade-offs between the accuracy and wall-clock time for various merging methods. For methods requiring hyperparameter tuning, e.g., WiSE-FT and Task Arithmetic, the wall-block time (logarithm of second) reflects a cumulative time for evaluations across all hyperparameters. For methods like `DaWin` or AdaMerging, we include the time required for additional workloads.

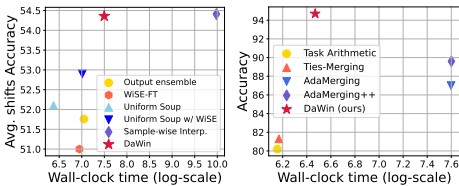

Figure 8: **Accuracy and wall-clock inference time with CLIP ViT-B/32**. We compare search-based methods, a test-time method (AdaMerging), and ours regarding the accuracy and inference time on robust fine-tuning (left) and multi-task learning evaluation on the SVHN dataset (right).

In both settings, `DaWin` shows favorable trade-offs that outperform the most efficient one in terms of accuracy while its runtime is far less than computation-heavier methods such as per sample interpolation without mixture modeling (Sample-wise Interp.) or AdaMerging. Thus, `DaWin` provides a high-performing merging solution that can be flexibly adapted considering a given cost budget.

**Theoretical analysis.** To understand the empirical success of `DaWin`, we present an analytic behavior of entropy-based dynamic weight interpolation in contrast to input-independent uniform weight interpolation, which yields a uniform interpolation coefficient (Wortsman et al., 2022a).

**Lemma 6.1 (Expert-biased weighting behavior of `DaWin`; Proof is deferred to Appendix A.4).** *Suppose we have $M$ different models $\{f(\cdot; \theta_j)\}_{j=1}^{M}$ parameterized by $\{\theta_j\}_{j=1}^{M}$ with a homogeneous architecture defined by $f(\cdot)$. Let $\lambda(x) = (\lambda_1(x), \ldots, \lambda_M(x))$ be the sample-wise interpolation coefficient vector given $x$, and $[f(x; \theta)]_c$ denotes the probability mass for class c. Then, we have*

$$\lambda_{j \in \mathcal{J}}(x) \geq \frac{1}{M} \quad \text{if } H(f(x; \theta_{j \in \mathcal{J}})) \leq H(f(x; \theta_{k \notin \mathcal{J}})) \ \forall j \text{ and } k,$$

$$\text{where } \mathcal{J} = \{i \mid \arg\max_c [f(x; \theta_i)]_c = y\}.$$

Lemma 6.1 implies that, under the entropy-dominancy assumption, `DaWin` always produces per-sample expert-biased coefficient vectors, which result in the interpolated models being biased towards true experts, i.e., models that produce correct prediction given $x$. This desirable behavior is aligned with the motivation of dynamic classifier selection discussed in Sec. 4.2, whereas `DaWin` conducts interpolation rather than selection. Meanwhile, as the number of models participating in interpolation increased, samples that at least one model correctly classifies also increased. Therefore, the coverage of `DaWin` for weighing the correct experts expands accordingly. This analysis endows a potential clue for remarkable gains observed in the multi-task setting, which conducts merging beyond two models (See Sec. 4.3).

**Conclusion.** This work has presented a novel training-free dynamic weight interpolation method, `DaWin`, that estimates the sample-wise model expertise with output entropy to produce reliable per-sample interpolation coefficients. We further proposed a mixture modeling approach, which greatly enhances computational efficiency. We then extensively evaluated `DaWin` with three different backbone models on two application scenarios, robust fine-tuning and multi-task learning, spanning 14 benchmark classification tasks. Regarding ID/OOD trade-offs in robust fine-tuning setup and overall accuracy in multi-task setup, `DaWin` consistently outperforms existing methods, implying its generality. This empirical success was further explained by discussing an analytic behavior of `DaWin`.

ACKNOWLEDGMENTS

We thank the researchers and interns at NAVER AI Lab – Byeongho Heo, Taekyung Kim, Sanghyuk Chun, Sehyun Kwon, Yong-Hyun Park, and Jaeyoo Park – for their invaluable feedback. Most of the experiments are based on the NAVER Smart Machine Learning (NSML) platform (Sung et al., 2017). We also appreciate on the constructive comments from Max Khanov, Hyeong Kyu Choi, and Seongheon Park at University of Wisconsin–Madison. This work was supported by the National Research Foundation of Korea (NRF) grant funded by the Korea government (MSIT) (RS-2024-00457216, NRF-RS-2024-00466956, FY2025).

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

# A  APPENDIX

We provide the following items in this Appendix:

- (A.1) Algorithm and additional details on `DaWin`
- (A.2) Additional details on the experiment setup
- (A.3) Additional empirical results
- (A.4) Missing proof
- (A.5) Limitation and future work

## A.1  ALGORITHM AND ADDITIONAL DETAILS ON `DaWin`

---

**Algorithm 1:** Procedure for training-free dynamic weight interpolation (**`DaWin`**)

---

**Input:** Test samples $X = \{x_i\}_{i=1}^N \in \mathbb{R}^{N \times D}$, models $f(\cdot; \theta_0)$ and $f(\cdot; \theta_1)$ of the same architecture, entropy function $H(\cdot)$, and the number of mixture component $K$

**for** $i = 1, \dots, N$ **do**

   $(H_{i,0}, H_{i,1}) \leftarrow (H(f(x_i; \theta_0)), H(f(x_i; \theta_1)))$  `// Per-sample model expertise`

   **if** *offset adjustment* **then**

      $\lambda(x_i) \leftarrow \frac{\exp(-H_{i,1}) + O(X)/T(X)}{\exp(-H_{i,0}) + \exp(-H_{i,1}) + O(X)}$

   **else**

      $\lambda(x_i) \leftarrow \frac{\exp(-H_{i,1})}{\exp(-H_{i,0}) + \exp(-H_{i,1})}$  `// Per-sample interp. coefficient`

   **end**

**end**

`bmm` $\leftarrow$ `BetaMixture(`$\{\lambda(x_i)\}_{i=1}^N$`;`$K$`).fit()`

$\{m_i\}_{i=1}^N \leftarrow$ `bmm.predict(`$X$`)`  `// Membership inference on test samples`

**for** $k = 0, \dots, K-1$ **do**

   $X_k \leftarrow X[\mathbb{I}(m_i == k), :]$

   $\lambda(X_k) \leftarrow$ `bmm[`$k$`].mean()`

   $\theta_{\lambda(X_k)} \leftarrow (1 - \lambda(X_k))\theta_0 + \lambda(X_k)\theta_1$  `// Cluster-wise dynamic interp.`

**end**

---

**Incorporating domain-wise expertise.**   While Eq. 1 yields sample-wise interpolation coefficients solely based on the per-sample expertise estimation, if we can access the whole test samples simultaneously, we can refine the coefficient computation by introducing domain-wise offset terms. These terms are automatically estimated by the per-domain entropy of each model. Given the significant performance improvements with domain-wise dynamic interpolation (*c.f.* Table 1), we expect domain-wise expertise per model can be a complimentary benefit to compute the interpolation coefficient. Therefore, we modify the sample-wise coefficient term in Eq. 1 as below:

$$\lambda(x) = \frac{\exp(-H(f(x; \theta_1))) + O(X)/T(X)}{\exp(-H(f(x; \theta_0))) + \exp(-H(f(x; \theta_1))) + O(X)}, \tag{2}$$

$$O(X) = \frac{1}{2}\left(\frac{\text{std}(H(f(X; \theta_0)))}{\text{mean}(H(f(X; \theta_0)))} + \frac{\text{std}(H(f(X; \theta_1)))}{\text{mean}(H(f(X; \theta_1)))}\right) \quad T(X) = \frac{H(f(X; \theta_0)) + H(f(X; \theta_1))}{H(f(X; \theta_0))},$$

where $O(X)$ and $T(X)$ denote the per-domain offset (formulated as an average coefficient of variation) and relative domain expertise terms, respectively. By incorporating domain-wise expertise, DaWIN offers more stable dynamic interpolation coefficients, correcting some of the inaccuracies in sample-wise estimation (see Table A). It is worth noting that these sample-wise and domain-wise expertise estimations **do not require any hyperparameters** for computing interpolation coefficients. This eliminates the need for intensive hyperparameter tuning that is often required in prior works (Wortsman et al., 2022b; Ilharco et al., 2023) to achieve the best performance.

**X-entropy ratio and likelihood ratio.**   Let $f(x; \theta) = p_\theta(y|x)$ be a classifier parameterized with $\theta$ that models the ground truth conditional probability distribution $p(y|x)$. Given that we observe one-hot encoded target labels so that $l(f(x; \theta), y) = -\log(p_\theta(y|x))$, then we can rewrite the oracle sample-wise coefficient in Sec 2.2 as follow, $\lambda^*(x) = p_{\theta_1}(y|x)/(p_{\theta_0}(y|x) + p_{\theta_1}(y|x))$.

**Details on Beta Mixture Model (BMM) with Expectation Maximization algorithm.** To enhance the inference-time efficiency of dynamic interpolation, we propose a mixture modeling approach over the estimated per-sample interpolation coefficients to reduce the number of interpolation operations from $N$ (entire test sample) to $K$ (pre-defined number of mixture components). Here, we elaborate on the detailed procedure of Beta Mixture Modeling[5].

We have coefficient estimates over $N$ test samples via Eq. 1 or Eq. 2 as $\{\lambda(x_i)\}_{i=1}^N$. Our goal is to model those coefficients with a mixture of $K$ Beta distributions as below:

$$\text{Beta}(\lambda(x_i); a_k, b_k) = \frac{\Gamma(a_k + b_k)}{\Gamma(a_k)\Gamma(b_k)}\lambda(x_i)^{a_k-1}(1 - \lambda(x_i))^{b_k-1} \tag{3}$$

$$p(\lambda(x_i)) = \sum_{k=1}^K \pi_k \text{Beta}(\lambda(x_i); a_k, b_k) \tag{4}$$

where $a_k > 0$ and $b_k > 0$ are the shape parameters of component $k$, $\Gamma(\cdot)$ and $\pi_k$ denote the Gamma function and mixing prior probabilities for each Beta component satisfying $\sum_{k=1}^K \pi_k = 1$ and $\pi_k \geq 0$. We first initialize the responsibilities $\{\gamma_{ik}\}$ by applying K-Means clustering to $\{\lambda(x_i)\}$ to assign the initial membership per each observation to one of $K$ components. We also initialize the parameter estimates of BMM as below:

$$\text{Mixing Priors}(\pi_k): \pi_k^{(0)} = \frac{N_k^{(0)}}{N}, \quad \text{where} \ \ N_k^{(0)} = \sum_{i=1}^N \gamma_{ik}^{(0)}. \tag{5}$$

$$\text{Shape Parameters}(a_k, b_k): a_k^{(0)} = C_k^{(0)} \times \bar{\lambda}_k^0 + \epsilon, \tag{6}$$

$$b_k^{(0)} = C_k^{(0)} \times (1 - \bar{\lambda}_k^{(0)}) + \epsilon \tag{7}$$

$$\text{where} \ \ \bar{\lambda}_k^{(0)} = \frac{1}{N_k^0}\sum_{i=1}^N \gamma_{ik}^{(0)}\lambda(x_i), \tag{8}$$

$$s_k^{2,(0)} = \frac{1}{N_k^0}\sum_{i=1}^N \gamma_{ik}^{(0)}(\lambda(x_i) - \bar{\lambda}_k^{(0)})^2, \tag{9}$$

$$C_k^{(0)} = \frac{\bar{\lambda}_k^0(1 - \bar{\lambda}_k^0)}{s_k^{2,(0)}} - 1, \tag{10}$$

where $\epsilon$ is a small positive constant to ensure numerical stability. Here, the initial shape parameters are estimated by *method-of-moments* (Pearson, 1936). Then, we conduct the expectation step (E-step) and the maximization step (M-step) alternatively until convergence to refine the parameter estimate as follows:

- **E-step**:
    - Compute log responsibilities $\ln\gamma_{ik}^{(t)}$.
    - Update responsibilities $\gamma_{ik}^{(t)} = \exp(\ln\gamma_{ik}^{(t)})$.
- **M-step**:
    - Update mixing priors $\pi_k^{(t+1)}$.
    - Update shape parameters $a_k^{(t+1)}, b_k^{(t+1)}$ using *method-of-moments* estimation.
- **Convergence Check**:
    - Compute log-likelihood $\mathcal{L}^{(t+1)}$, where $\mathcal{L}^{(t)} = \sum_{i=1}^N \ln p(\lambda(x_i))$
    - If $|\mathcal{L}^{(t+1)} - \mathcal{L}^{(t)}| < $ tolerence, stop the iterations.

Then, we get the estimated parameter $\Theta = \{\pi_k, a_k, b_k\}_{k=1}^K$ of BMM to infer per-sample weight interpolation coefficients.

---

[5]The same procedure is adopted for the Dirichlet Mixture Model in multi-task learning scenario by modifying the probability density function from Beta distribution to Dirichlet distribution.

## A.2 Additional details on the experiment setup

In this section, we provide extended details for task definition, baseline methods, and implementation details. Some contents might be duplicated from the main paper.

### A.2.1 Tasks and Datasets

We validate DaWin on two scenarios: (1) **robust fine-tuning** (Wortsman et al., 2022b) and (2) **multi-task learning** (Ilharco et al., 2022) with focusing on the top-1 classification accuracy (Acc). Following robust fine-tuning literature (Wortsman et al., 2022b; Kumar et al., 2022), we use ImageNet-1K (Russakovsky et al., 2015) and its five variants, ImageNet-V2 (Recht et al., 2019), a post-decade reproduced version of the original ImageNet test set by following the dataset generating process of ImageNet, ImageNet-R (Hendrycks et al., 2021a), a rendition-specific collection of 200 ImageNet classes, ImageNet-A (Hendrycks et al., 2021b), an actual examples from ImageNet test set misclassified by a ResNet-50 model over 200 ImageNet classes, ImageNet-Sketch (Wang et al., 2019), a sketch-specific collection of 1000 ImageNet classes, and ObjectNet (Barbu et al., 2019) for evaluating robustness under distribution shifts. For multi-task learning, we follow the standard evaluation protocol (Ilharco et al., 2022; Yang et al., 2024b) using eight benchmark datasets from the optical character images, traffic signs, scenery or satellite imagery, and fine-grain categorization over cars and texture: SUN397 (Xiao et al., 2016), Cars (Krause et al., 2013), RESISC45 (Cheng et al., 2017), EuroSAT (Helber et al., 2019), SVHN (Yuval, 2011), GTSRB (Stallkamp et al., 2011), MNIST (LeCun, 1998), and DTD (Cimpoi et al., 2014).

### A.2.2 Models and Baselines

For robust fine-tuning, we adopt CLIP ViT-{B/32, B/16, L/14} (Radford et al., 2021) as zero-shot (ZS) models to ensure a fair comparison with (Wortsman et al., 2022b;a; Ilharco et al., 2022; Yang et al., 2024b), and fine-tuned (FT) checkpoints for each CLIP ViT backbone from Jang et al. (2024). For the weight interpolation baseline methods, we include WiSE-FT (Wortsman et al., 2022b), which conducts a weight interpolation between a pre-trained model and a fine-tuned model given a single pre-defined interpolation coefficient, Model Soup (Wortsman et al., 2022a), which conducts averaging of all models' weights (Uniform Soup) or greedily selected models' weights (Greedy Soup) trained with different training hyperparameter configurations, and Model Stock (Jang et al., 2024), iteratively interpolate a pre-trained model with few fine-tuning model based on the cosine distance between pre-trained model weight and the fine-tuning model weights, along with the traditional output ensemble method. In addition, we consider several state-of-the-art robust fine-tuning methods such as LP-FT (Kumar et al., 2022), a two-stage method to avoid pre-trained feature distortion, CAR-FT (Mao et al., 2024), a regularized fine-tuning method leveraging context-aware prompt, FLYP (Goyal et al., 2023), a contrastive learning based fine-tuning method, Lipsum-FT (Nam et al., 2024), a regularized fine-tuning method motivated by energy score gap between zero-shot and fine-tuned models, and CaRot (Oh et al., 2024), a theory-inspired singular value regularization method.

For multi-task learning, we use CLIP ViT-B/32 as our backbone and consider the model merging baselines as follows: a simple weight averaging (Ilharco et al., 2022), Fisher Merging (Matena & Raffel, 2022), a Fisher information metric-based weighted averaging method, RegMean (Jin et al., 2023), an averaging method that minimizes $L_2$ distance between the averaged weight and individual weights, Task Arithmetic (Ilharco et al., 2023), a method perform arithmetic across task vectors rather the original weight vector itself which are produced by the subtractions between a pre-trained model weight and individual fine-tuned model weights, Ties-Merging (Yadav et al., 2023), a post-hoc weight refinement method mitigating conflicts between task vectors, AdaMerging (Yang et al., 2024b) and Pareto Merging (Chen & Kwok, 2024) methods those driving additional optimization procedure to a global interpolation coefficient and the conditional coefficient generation models that trained to minimize entropy over entire test sample.

### A.2.3 Implementation Details

For fine-tuning CLIPs on ImageNet, Wortsman et al. (2022a) and its successor (Jang et al., 2024) conducted multiple training with different training configurations such as data augmentation, learning rate, weight decay, and random initialization seeds given fixed epochs (16) and batch size (512). Here, we use the best model weight provided by the authors of Jang et al. (2024) per each backbone.

For fine-tuning weights of CLIP on multi-task learning setup, we adopt the official checkpoints from Ilharco et al. (2023) on the eight datasets.

On DaWin's evaluation, we first get the entropy of batch test samples from the interpolation candidate models[6] (wherein temperature scaling (Guo et al., 2017) is applied in the robust fine-tuning setup with ID validation set), then we compute interpolation coefficients by building a softmax-like model expertise ratio term with exponentiated negative entropy of each model. We further perform the mixture modeling over the batch test samples and finally conduct dynamic model merging with interpolation coefficients corresponding to estimated membership from the fitted mixture model to obtain the prediction per sample. About the Beta (on robust fine-tuning setup) and Dirichlet (on multi-tasks learning setup) mixture modeling on interpolation coefficients for DaWin, we set $K$ to 3, 5, 2 for ViT-{B/32, B/16, L/14} in the robust fine-tuning and $K = 1$ in the multi-task setups. Unless otherwise mentioned, we adopt the offset adjustment term (Eq. 2) by default, assuming that the entire test samples per task are available and fit the Beta (and Dirichlet) mixture model on the entire coefficients per task, likewise assumption of Yang et al. (2024b;a); Chen & Kwok (2024).

## A.3 ADDITIONAL EMPIRICAL RESULTS

Table A: **Ablation study on offset adjustment**. We validate the effect of using the offset adjustment term on ImageNet ID and OOD accuracy.

| Model | Offset Adjustment | ID | Avg. Acc on OOD |
|---|---|---|---|
| ViT-B/32 | - | 78.3 | 54.1 |
|  | ✓ | **78.7** | **54.4** |
| ViT-B/16 | - | 83.1 | 66.6 |
|  | ✓ | **83.4** | **66.9** |
| ViT-L/14 | - | 86.7 | 75.9 |
|  | ✓ | **86.9** | **76.0** |

We ablate the offset adjustment and expertise metric to investigate the effectiveness of the design choices of each component. Firstly, as we can see in Table A, offset adjustment consistently boosts the ID and OOD accuracy across all cases, which supports the use of domain-wise relative expertise (average entropy over all test samples) to enhance sample-wise expertise estimation. To secure

Table B: **Sensitivity analysis on sample size.** We report DaWin's ImageNet and its OOD variants' accuracy of CLIP ViT-B/32 under varying sample wise for fitting Beta Mixture Model (where the number of mixture components $K$=3). DaWin shows robustness against varying sample sizes.

| Sample size | ImageNet Accuracy | Avg. Acc on OOD |
|---|---|---|
| 32 | 78.55 | 54.30 |
| 64 | 78.69 | 54.27 |
| 128 | 78.71 | 54.27 |
| 256 | 78.70 | 54.28 |
| 512 | 78.70 | 54.25 |
| 1024 | 78.71 | 54.24 |
| 2048 | 78.67 | 54.26 |
| $N$ | 78.70 | 54.36 |

inference time efficiency, we adopt the Beta mixture modeling approach on the batch-wise DaWin's coefficients. The fitness of the Beta mixture model may be improved as the sample size is increased, whereas a smaller sample size enables more granular interpolations to be applied. Therefore, we evaluate the performance of DaWin under varying sample sizes. Table B presents the ID and OOD performance of DaWin on the ImageNet distribution shift benchmark for the CLIP ViT-B/32 backbone model. DaWin shows strong robustness against varying sample sizes.

---

[6]Candidate models are constituted with the pre-trained and fine-tuned models for robust fine-tuning setting, the task-specific eight fine-tuned models for multi-task learning setting.

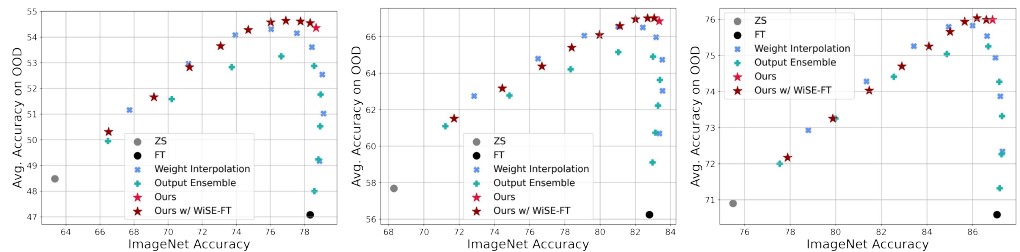

Figure A: **ID and OOD trade-off analysis**. We visualize performance trade-offs regarding ID and OOD classification accuracy on the ImageNet distribution shift benchmarks with CLIP ViT-{B/32, B/16, L/14} from left to right. Interpolation coefficients are swept over $\{0.1, 0.2, ..., 0.9\}$.

In Figure A, we present the results of `DaWin` with WiSE-FT as a plug-in augmentation on the robust fine-tuning setup. We multiply our estimated coefficients from Beta mixture by a scalar coefficient $\alpha$, e.g., $\lambda_{wise}(x) = \lambda(x) \times \alpha$. Although it brings slight benefits in the case of CLIP ViT-B/32, the WiSE-FT interpolation trace becomes almost a line on the ViT-B/16 and ViT-L/14 cases. This implies that `DaWin` already achieves performance beyond the Pareto-optimal trade-offs and cannot be further improved by WiSE-FT, given models $f(\cdot; \theta_0)$ and $f(\cdot; \theta_1)$. Meanwhile, Figure B reveals that `DaWin` produces larger $\lambda(x)$ for samples that are hard to recognize the target object due to overwhelming background semantics, which the pre-trained model may wrongly pay attention to.

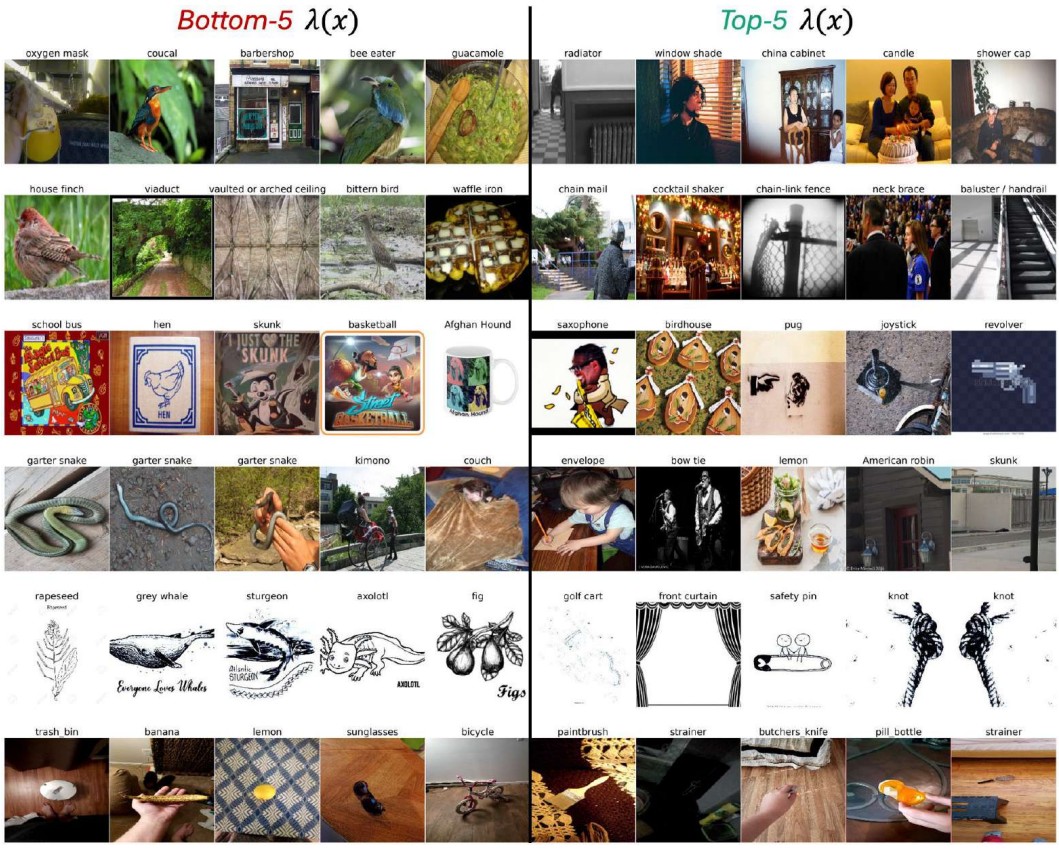

Figure B: **DaWIN's bottom-5 and top-5 estimated coefficient analysis**. We visualize the images with labels corresponding to bottom-5 and top-5 sample-wise interpolation coefficients estimated by DaWIN of pre-trained and ImageNet fine-tuned CLIP ViT-B/32. Each row denotes the actual test samples from ImageNet, ImageNet-V2, ImageNet-R, ImageNet-A, ImageNet-Sketch, and ObjectNet. We see that the images corresponding to coefficients lean towards the fine-tuned model, which typically contains multiple semantics, and the object corresponding to the ground truth label is overwhelmed by other semantics.

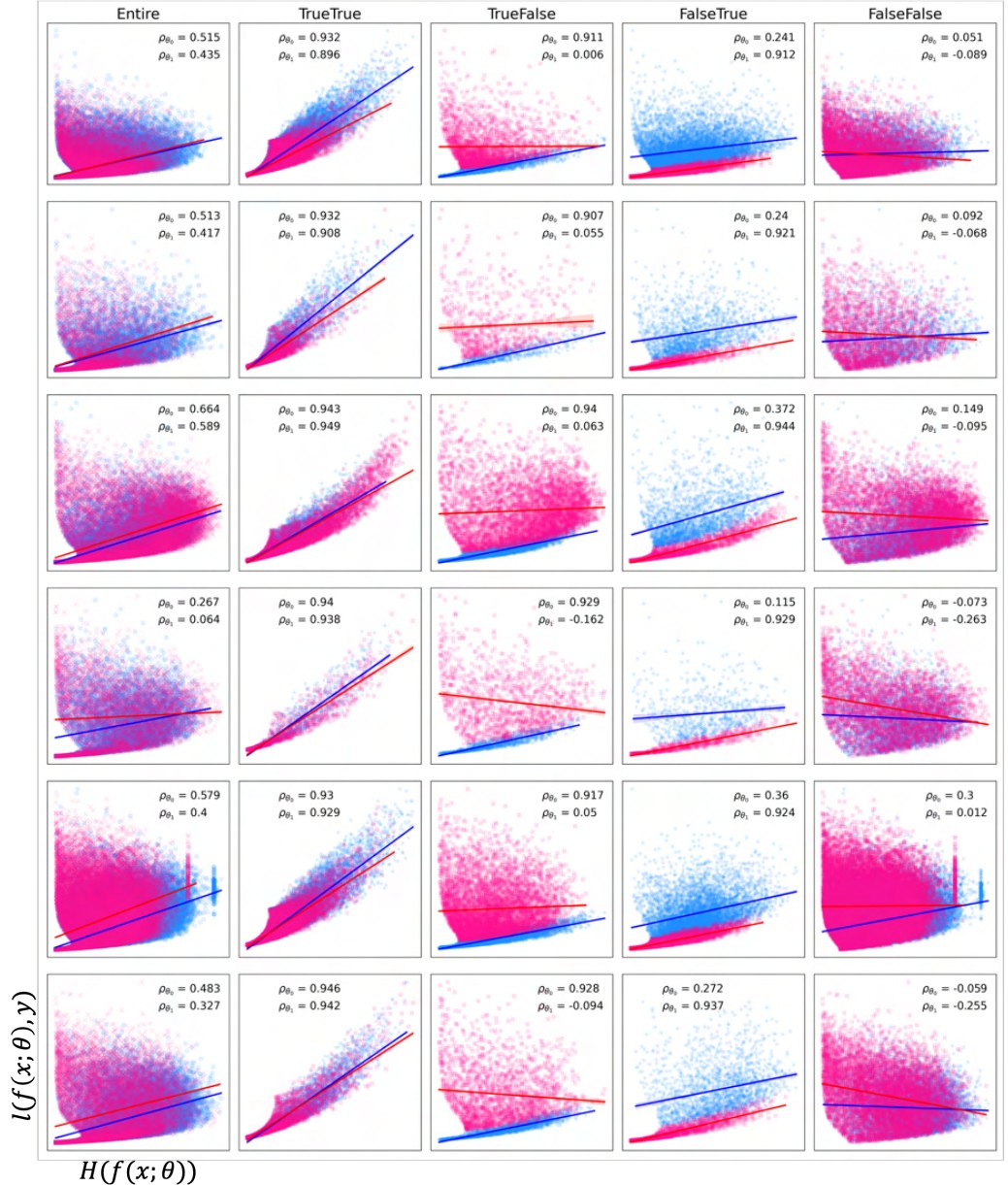

Figure C: **Correlation between entropy and X-entropy**. We split each testset into four subsets (`TrueTrue`, `TrueFalse`, `FalseTrue`, `FalseFalse`) based on the correctness of the two models' predictions. On each data split and the entire dataset, we visualize scatter plots of entropy (x-axis) and cross-entropy (y-axis) computed by two models (zero-shot and fine-tuned) with corresponding Pearson correlation coefficients per model. Each row from top to bottom shows results from ImageNet, ImageNetV2, ImageNetR, ImageNetA, ImageNetSketch, and ObjectNet.

`DaWin` adopts entropy as a proxy of X-entropy to estimate the model expertise without access to the true target label. Figure C presents the correlation between entropy and X-entropy of the pre-trained CLIP ViT-B/32 $f(\cdot; \theta_0)$ and ImageNet fine-tuned counterpart $f(\cdot; \theta_1)$. Results indicate that the model producing correct predictions holds a strong correlation between entropy and X-entropy while the model failing to correctly predict test samples shows bad or no correlation. However, if at least one model success in making a correct prediction in a given sample $x$, and the entropy of the correct predictor may be smaller than that of another model, thereby `DaWin` would be likely to produce $\lambda(x)$ biased towards the correct predictor's weight (See Lemma 6.1).

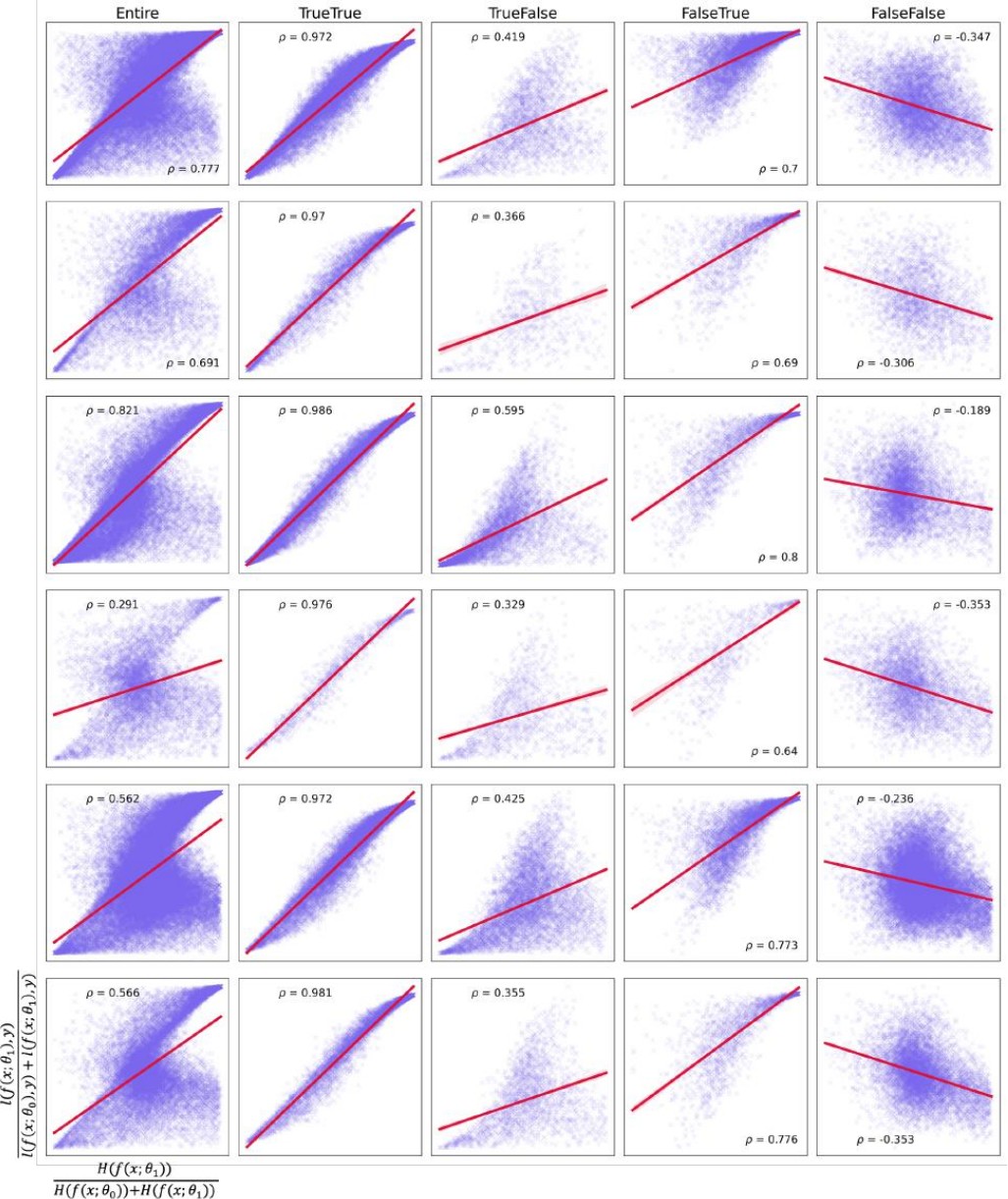

Figure D: **Correlation between entropy ratio and X-Entropy ratio**. We split each testset into four subsets (TrueTrue, TrueFalse, FalseTrue, FalseFalse) based on the correctness of the two models' predictions. On each split and the entire dataset, we visualize scatter plots of entropy ratio (x-axis) and cross-entropy ratio (y-axis) computed by two models (zero-shot and fine-tuned) with corresponding Pearson correlation coefficients. Each row from top to bottom shows results from ImageNet, ImageNetV2, ImageNetR, ImageNetA, ImageNetSketch, and ObjectNet.

In Figure D, we provide the extended results of Figure 2, showing the correlations between entropy ratio and X-entropy ratio, on the whole evaluation datasets of robust fine-tuning setup. The entropy ratio approximates the X-entropy ratio overall across datasets and sub-populations of each dataset, even though for the most challenging OOD, i.e., ImageNet-A, which is constructed with natural adversarial examples, there is a weak-yet-non-trivial correlation (Schober et al., 2018) between entropy and X-entropy. Moreover, we note that weight interpolation (or output ensemble) has an advantage that elicits the correct prediction by modifying the relative feature importance (Yong et al., 2024) even though two individual models fail to produce correct predictions (i.e., in the FalseFalse case), thereby expected to outperform the model selection method (See Table 5).

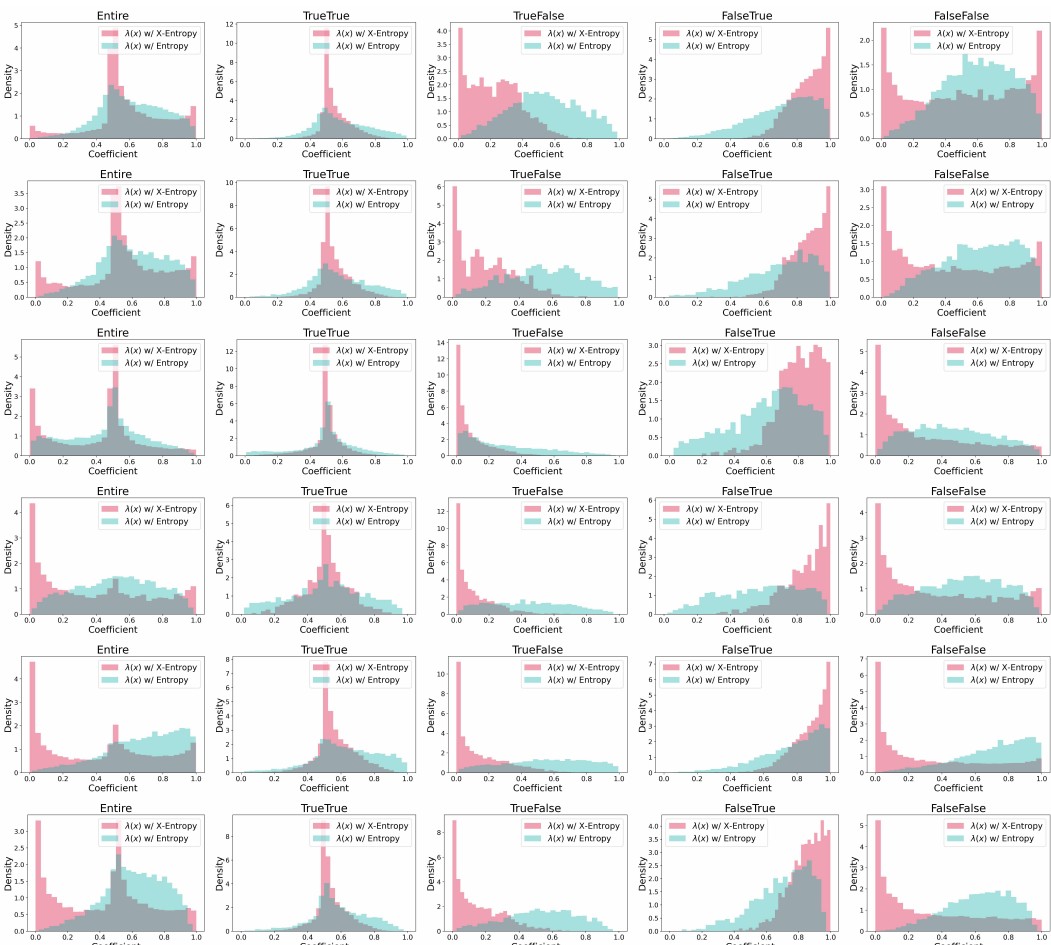

Figure E: **Estimated coefficient density analysis on ImageNet distribution shift benchmarks with CLIP ViT-B/32**. We visualize histograms of estimated sample-wise interpolation coefficients by the expertise ratio, which is computed using X-entropy (Oracle; we can not access it in reality) and entropy as expertise metrics. Each column denotes the splits {Entire, TrueTrue, TrueFalse, FalseTrue, FalseFlase} of the test set, those are categorized by the correctness of zero-shot CLIP, and ImageNet fine-tuned CLIP's predictions. Each low denotes the result of the test set: from ImageNet, ImageNet-V2, ImageNet-R, ImageNet-A, ImageNet-Sketch, and ObjectNet. Entropy-based coefficient estimation shows remarkably good fitness in the TrueTrue, FalseTrue splits, and produces left-skewed distribution in the case of TrueFalse, which is desired to construct the interpolated model biased towards zero-shot model weight. Overall, except the FalseFalse case, the entropy-based coefficient estimation provides reasonable alternatives to X-entropy-based oracle coefficients.

We visualize the histogram of interpolation coefficients computed by entropy and X-entropy ratio in Figure E. While the entropy-based coefficients quite diverge from the oracle X-entropy-based coefficients in some cases (e.g., TrueFalse and FalseFalse of ImageNet-Sketch and ObjectNet), the overall distributions of entropy-based coefficients show good fitness to X-entropy-based coefficients across datasets. This result supports using entropy as a proxy of X-entropy to estimate model expertise given unlabeled test-time input to determine the per-sample interpolation coefficient.

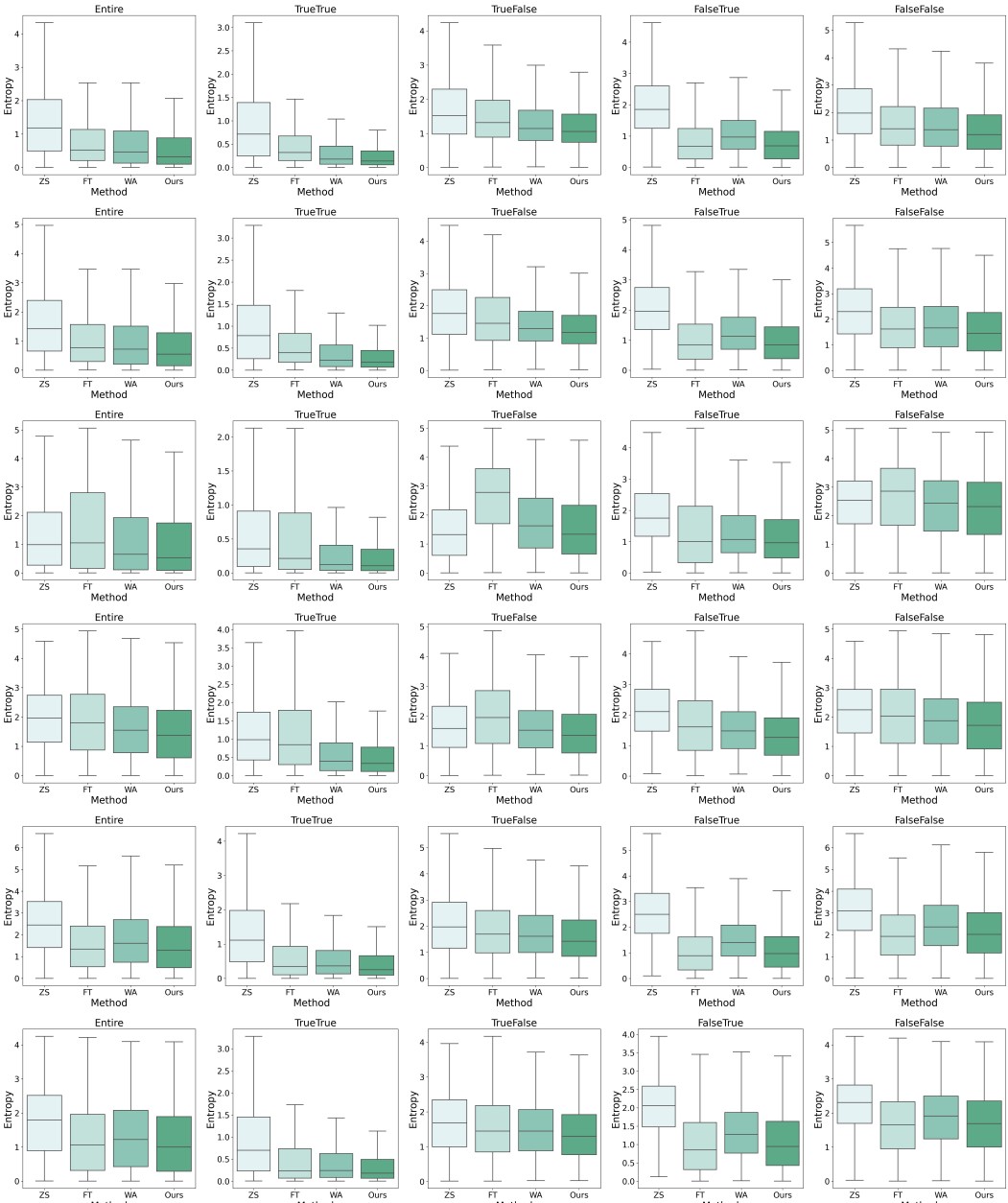

Figure F: **Entropy comparison on ImageNet distribution shift └ with CLIP ViT-B/32**. Across IN, IN-V2, IN-R, IN-A, IN-S, and ObjNet from top to bottom, we visualize the final output entropy from each model on the entire dataset and four different splits based on the correctness of zero-shot (ZS) and fine-tuned (FT) models. Compared with individual models, weight averaging (WA) induces lower entropy overall, and our `DaWin` achieves the lowest entropy across all splits.

In Figure F, we provide the extended results of Figure 6, showing the average entropy over test samples, on the whole evaluation datasets of robust fine-tuning setup. In almost all of cases, weight interpolation achieves smaller entropy than individual models, and `DaWin` achieves the smallest entropy. This supports our *sample-wise entropy valley* hypothesis in Section 3.2 and helps us to understand `DaWin`'s great performance gains.

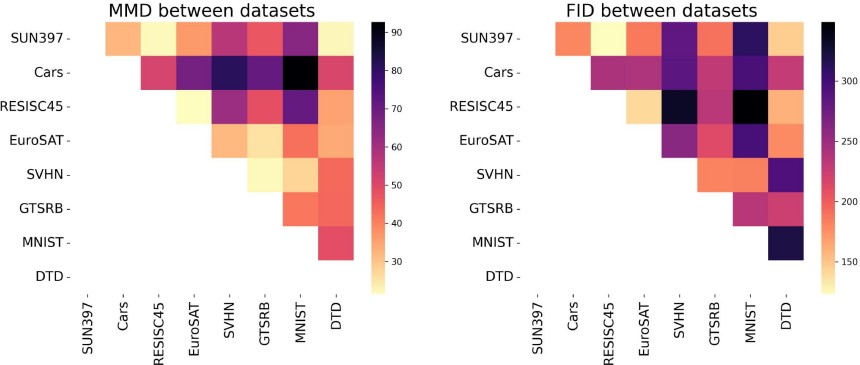

Figure G: **Distance between datasets**. We visualize the mean maximum discrepancy (MMD) and Frechet inception distance across eight datasets used in multi-task learning benchmarks in the left and right panels, respectively. To compute MMD and FID, we adopt the pre-trained OpenAI CLIP ViT-B/32 and ImageNet pre-trained Inception V3 models as feature extractors respectively.

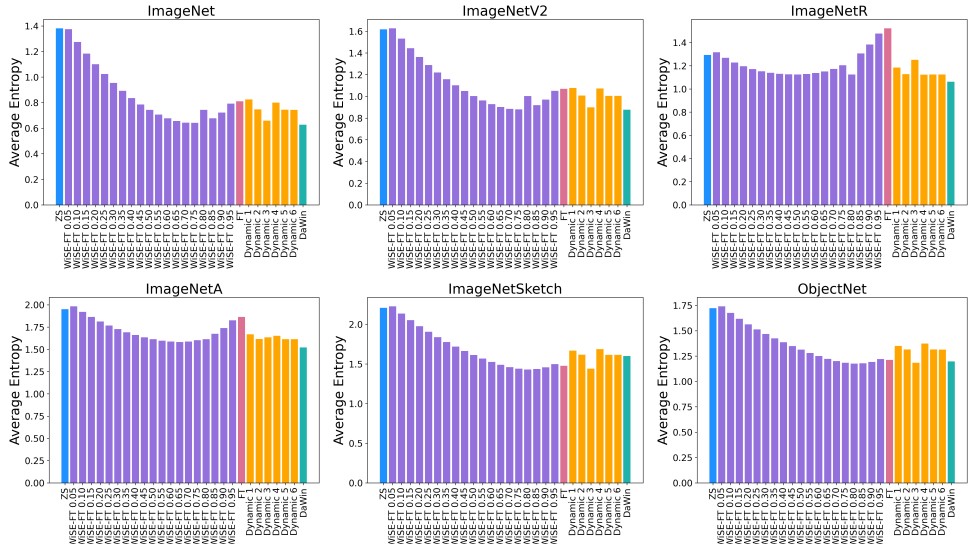

Figure H: **Empirical validation on the sample-wise entropy valley hypothesis**. We visualize the average entropy for each dataset across various methods including single model evaluations (ZS and FT), static weight interpolation (WiSE-FT 0.05, ..., 0.95), some dynamic weight interpolation methods ($U(0,1)$, $N(0.5, 0.2^2)$, $N(0.8, 0.1^2)$, Max logit ratio, Confidence ratio, and Confidence difference ratio), and our DaWin method.

In Figure G, we visualize the distance between eight datasets used in the multi-task learning setups. The computed cross-dataset distances are matched with the average coefficients derived from DaWin in Figure 7 to some extent.

Similar to the linear mode connectivity phenomenon that prevails between pre-trained and fine-tuned checkpoints of foundation models (Wortsman et al., 2022b; Ramé et al., 2023) which indicates the interpolations between two checkpoints do not increase the expected loss (e.g., X-entropy) on downstream, in Section 3.2, we hypothesized the expected entropy of model output may also show similar convexity trend over interpolations between pre-trained and fine-tuned model checkpoints. In Figure H, we see that this claim empirically holds for the six datasets including ID (ImageNet) and OOD (its five variances), i.e., there exists an interpolated model that achieves smaller entropy than individual models (ZS and FT). Besides, we further observed that there exists a per-sample dynamic interpolation method that achieves much smaller entropy compared to the static interpolation method. Therefore, our two statements in the sample-wise entropy valley hypothesis show empirical evidence.

Table C: **Optimality check for $\lambda^*(x)$.** We provide mean squared error (MSE) between the optimally estimated per-sample interpolation coefficients and our X-entropy ratio-based estimation in Section 2.2. We estimate the optimal coefficient per sample by conducting a finer-grained grid search for the coefficients, e.g., $\{0.05, 0.10, ..., 0.90, 0.95\}$ per sample to minimize the X-entropy loss of interpolated model given a sample $x$. As baseline coefficients, $U(0, 1)$ and WiSE-FT constant (0.8) denote per sample random uniform variable and a constant scalar 0.8 for all samples respectively.

| - | ImageNet | ImageNetV2 | ImageNetR | ImageNetA | ImageNetSketch | ObjectNet |
|---|---|---|---|---|---|---|
| Std. of per-sample optimal coefficients $\lambda^*(x)$ | 0.2827 | 0.3068 | 0.3237 | 0.3287 | 0.3413 | 0.3438 |
| MSE between $\lambda^*(x)$ and $U(0, 1)$ | 0.1797 | 0.1823 | 0.1869 | 0.1976 | 0.1980 | 0.2013 |
| MSE between $\lambda^*(x)$ and WiSE-FT constant | 0.1084 | 0.1390 | 0.2081 | 0.2427 | 0.2036 | 0.2131 |
| MSE between $\lambda^*(x)$ and X-entropy ratio (ours) | 0.0382 | 0.1559 | 0.0479 | 0.0364 | 0.0400 | 0.0440 |

In Section 2.2, we adopted the negative X-entropy as an oracle model expertise metric and used the exponentiated negative X-entropy ratio as our desired per-sample interpolation coefficients. Although the exponentiated negative X-entropy ratio $\lambda^*(x)$ has a nice interpretation connected to the density estimation (Sec A.1), it is not guaranteed that the $\lambda^*(x)$'s optimality in the context of weight interpolation. To investigate how close our $\lambda^*(x)$ are to the true optimal per-sample interpolation coefficients that minimize the X-entropy losses of interpolation model given samples, we measure mean squared error (MSE) between ours and optimally estimated per-sample coefficients (we estimate these by conducting grid search for interpolation coefficient over $\{0.05, 0.10, ..., 0.90, 0.95\}$ per sample). Table C presents the results indicating that our $\lambda^*(x)$ are generally much closer to the optimally estimated coefficient compared with random uniform variable or tuned WiSE-FT constant coefficient. Given that the standard deviation of oracle coefficients are about from 0.28 to 0.34, the closeness between optimal and X-entropy ratio-based coefficients is remarkable.

Table D: **Alternative per-sample interpolation coefficients.** We provide results from some different design choices for per-sample dynamic interpolations (Beta Mixture Modeling with $K$=3 applied for efficient inference) on ImageNet distribution shift benchmarks. Although other data-independent or data-dependent methods struggle to strike a good balance between ID and OOD performances, `DaWin` shows outstanding performance among considered candidates demonstrating its effectiveness well-grounded with its unique motivation.

| Category | Coefficient per sample | ID Acc | OOD Acc Avg. |
|---|---|---|---|
| No interpolation (FT) | - | 78.4 | 47.1 |
| Static interpolation | 0.8 (WiSE-FT) | 79.1 | 51.0 |
| Data-independent | $U(0, 1)$ | 75.9 | 52.9 |
| Data-independent | $N(0.5, 0.2^2)$ | 77.5 | 54.1 |
| Data-independent | $N(0.8, 0.1^2)$ | 79.1 | 51.1 |
| Data-dependent | Max Logit Ratio | 76.6 | 54.3 |
| Data-dependent | Confidence Diff. Ratio | 77.5 | 54.2 |
| Data-dependent | Confidence Ratio | 77.5 | 54.2 |
| Data-dependent | Entropy Ratio (ours) | 78.7 | 54.4 |

In Table D, we provide some alternative design choices for functions that estimate per-sample interpolation coefficients from two categories, data-independent and data-dependent. We see that `DaWin` outperforms other baseline methods indicating our careful design choice of `DaWin` motivated by a pilot study has non-trivial benefits rather than other heuristic alternatives.

Table E: **Performance of DaWin w/ and w/o confidence calibration on ImageNet (ID) and its variants (OOD).** Here, we adopt temperature scaling on ID validation set to calibrate individual models' outputs.

| Method | ID Acc | OOD Acc |
|---|---|---|
| WiSE-FT | 79.1 | 51.0 |
| DaWin (default) | 78.7 | 54.4 |
| DaWin w/ bad calibration | 76.6 | 54.0 |

Meanwhile, `DaWin` adopts the entropy of output probability distributions of individual models. As mentioned in Section A.2, we apply temperature scaling to individual merging candidate models

to get calibrated outputs from them. The natural question is "how does the uncertainty calibration of individual models affect the final performance of DaWin?". We adopt temperature scaling on the ID validation set by default to calibrate individual models in the robust fine-tuning setup, so we now ablate the temperature scaling in Table E. We can see that DaWin without temperature scaling (denoted as bad calibration) somewhat compromises ID performance. However, it still significantly outperforms the static merging method on OOD performance which demonstrates the effectiveness of our method even without delicate tuning.

Table F: **Runtime, FLOPs, and Peak memory allocation comparison between static (WiSE-FT) and dynamic (DaWin) weight interpolation for evaluation on ImageNet variants.** Here, we use ViT-B/32 backbone model on NVIDIA A100 GPU(s). DaWin (parallel) conducts the expertise computation process in parallel (i.e., forward evaluations for pre-trained and fine-tuned models are conducted simultaneously) with multiple GPUs for faster inference.

| Method | Total Runtime (sec) | FLOPs per sample | Peak memory (MB) per sample | OOD Avg. Acc |
|---|---|---|---|---|
| WiSE-FT | 1040 | 4.4139 | 454 | 51.0 |
| DaWin | 1802 | 13.2452 | 1338 | 54.4 |
| DaWin (parallel) | 1341 | 8.8313 | 1338 | 54.4 |

Along with Figure 8, to further investigate the trade-offs between DaWin and a simpler method, we provide evaluation results with additional metrics (FLOPs and peak memory allocation) and variants of DaWin in Table F. Here, DaWin (parallel) denotes the setting where we parallelize the evaluations of individual models during the expertise estimation phase before conducting the interpolation[7]. DaWin shows three times FLOPs than WiSE-FT, but the actual runtime is less than twice that of WiSE-FT because DaWin does not require intensive hyperparameter tuning, unlike WiSE-FT. Besides, the computational complexity can be further reduced by adopting the parallel inference pipeline for the model expertise estimation phase of DaWin.

Table G: **Fine-grained analysis on interpolation candidates.** Model Pool denotes the candidate models (fine-tuned on those datasets) we use to build the interpolated model. We simulate some scenarios where we use relevant expert models only or mixed candidates of experts and non-experts to get the interpolated model. Relevant datasets are determined based on task and semantic similarity between datasets.

| Scenario | Model Pool | Evaluation data | Task Arithmetic | AdaMerging++ | DaWin |
|---|---|---|---|---|---|
| Experts only | SVHN, MNIST | SVHN | 87.5 | **94.1** | 90.5 |
| Mixed | SVHN, MNIST, EuroSAT | SVHN | 84.4 | 93.7 | **93.8** |
| Mixed | SVHN, MNIST, EuroSAT, RESISC45 | SVHN | 81.5 | 93.4 | **94.8** |
| Experts only | EuroSAT, RESISC45 | EuroSAT | 96.4 | 98.1 | **98.6** |
| Mixed | SVHN, MNIST, EuroSAT | EuroSAT | 86.9 | 97.7 | **99.6** |
| Mixed | SVHN, MNIST, EuroSAT, RESISC45 | EuroSAT | 89.6 | 97.7 | **99.7** |

Including our method, the success of model merging methods (also traditional ensemble methods) depends on the relation between the training datasets of merging candidate models and the evaluation dataset. In Table G, we systematically analyze the effect of the relation between train and evaluation datasets on the performance of the merged method by constructing a model pool for merging based on the distance between train datasets (where the candidate models are trained) and test datasets. Please refer to Figure G and Figure 7 to check the quantities of similarity between datasets.

We simulate scenarios where all models are relevant to solve the downstream task, or some models are relevant but others are not to solve the downstream task. Although the baseline methods perform well when we have expert models trained on datasets that are relevant to the evaluation dataset, they suffer from performance degeneration as the non-expert models are included in the merging pool. This indicates that there are some interferences between models (and train corresponding datasets) that hurt post-merging performance if we inappropriately determine the interpolation coefficient. Meanwhile, **DaWin takes benefits from even non-expert models by leveraging the entropy ratio-based weighting strategy**, and the performance is improved as more models participate in merging.

---

[7]This treatment is orthogonal to the commonly used distributed data-parallel inference mode and is only applicable to DaWin.

## A.4 MISSING PROOF

In Sec. 6, we provided an analytic behavior of `DaWin`, which generated true-expert biased interpolation weight vectors as below:

**Lemma A.1** (**Restatement of expert-biased weighting behavior of `DaWin`**). *Suppose we have $M$ different models $\{f(\cdot; \theta_j)\}_{j=1}^{M}$ parameterized by $\{\theta_j\}_{j=1}^{M}$ with a homogeneous architecture defined by $f(\cdot)$. Let $\lambda(x) = (\lambda_1(x), \ldots, \lambda_M(x))$ be the sample-wise interpolation coefficient vector given $x$, and $[f(x; \theta)]_c$ denotes the probability mass for class c. Then, we have*

$$\lambda_{j \in \mathcal{J}}(x) \geq \frac{1}{M} \quad if \ \ H(f(x; \theta_{j \in \mathcal{J}})) \leq H(f(x; \theta_{k \notin \mathcal{J}})) \ \ \forall j \ and \ k,$$
$$where \ \ \mathcal{J} = \{i | \arg\max_c [f(x; \theta_i)]_c = y\}.$$

*Proof.* The proof is very straightforward, given the assumption and definitions of problem setup.

$$H(f(x; \theta_{j \in \mathcal{J}})) \leq H(f(x; \theta_{k \notin \mathcal{J}})) \quad \text{(By assumption)}$$
$$\exp(-H(f(x; \theta_{j \in \mathcal{J}}))) \geq \exp(-H(f(x; \theta_{k \notin \mathcal{J}})))$$
$$\lambda_{j \in \mathcal{J}}(x) \geq \lambda_{k \notin \mathcal{J}}(x) \quad \text{(By definition of } \lambda(x))$$
$$\lambda_{j \in \mathcal{J}}(x) \geq \frac{1}{M} \quad \text{(Given that } \sum_{j=1}^{M} \lambda_j(x) = 1)$$

$\square$

## A.5 LIMITATION AND FUTURE WORK

By following previous works (Wortsman et al., 2022b; Ilharco et al., 2023; Yang et al., 2024b), we limited our validation scope to an image classification of fully fine-tuned visual foundation models with restricted scale and architecture, e.g., CLIP ViT-{B/32, B/16, L/14}. Exploring `DaWin` on diverse model architecture (Liu et al., 2022; Gu & Dao, 2023; Liu et al., 2024), large-scale modeling setup (Dehghani et al., 2023), large language models (Yu et al., 2024; Goddard et al., 2024; Rame et al., 2023), multimodal generative models (Liu et al., 2023; Biggs et al., 2024; Nair et al., 2024), continual adaptation scenario (Marczak et al., 2024), and parameter-efficient tuning regime (Li & Liang, 2021; Hu et al., 2021; Chronopoulou et al., 2023) can be exciting future work directions.

Meanwhile, the increased amount of computation during inference time is another limitation of `DaWin`. However, we note that scaling test-time computation can be a cost-effective solution for foundation models to address challenging tasks and data in the wild (Snell et al., 2024; Jaech et al., 2024; Muennighoff et al., 2025). Given the observations that modern multimodal foundation models still struggle with *out-of-distribution* query (OpenAI, 2024; Zhang et al., 2024; Oh et al., 2025), improving the robustness of foundation models by allocating more budgets for test-time computation through dynamic merging, such as `DaWin`, is worth to be investigated further.

