# OpenReview forum: "DaWin: Training-free Dynamic Weight Interpolation for Robust Adaptation"
_ICLR.cc/2025/Conference — ICLR 2025 Poster_

### Official Review · Reviewer_yGMX · 2024-10-27

**Soundness:** 3
**Presentation:** 3
**Contribution:** 3
**Rating:** 6
**Confidence:** 3

**Summary:**

The paper titled "DaWin: Training-Free Dynamic Weight Interpolation for Robust Adaptation" introduces DaWin, a novel method for dynamically merging model weights without additional training. Traditional weight interpolation methods, such as WiSE-FT, use static coefficients for merging models, limiting their effectiveness when models exhibit different levels of expertise across samples. DaWin addresses this limitation by proposing sample-wise dynamic weight interpolation, utilizing entropy-based measures to adjust interpolation coefficients in real-time. DaWin achieves state-of-the-art performance on both robust fine-tuning tasks (ImageNet benchmarks) and multi-task learning (e.g., EuroSAT, MNIST) with minimal computational overhead.

**Strengths:**

1. Training-free approach: DaWin eliminates the need for additional model training, providing a plug-and-play solution.
2. Entropy-based dynamic interpolation: This novel use of entropy helps improve sample-wise performance across diverse tasks.
3. State-of-the-art results: DaWin outperforms WiSE-FT and other interpolation methods on ImageNet and multi-task benchmarks with minimal computational overhead.
4. Computational efficiency: The use of mixture modeling reduces the inference cost, making DaWin scalable to large datasets.

**Weaknesses:**

1. **Incremental novelty**: While the method provides refinements over static interpolation, it shares similarities with existing methods (e.g., AdaMerging), which reduces its overall originality.
2. Over-reliance on entropy: The assumption that entropy reliably measures model expertise across all scenarios is not thoroughly validated, and alternative metrics could be explored further.
3. Trade-offs not fully explored: The paper does not sufficiently address the performance trade-offs between per-sample interpolation and simpler methods like weight averaging.

**Questions:**

1. How well does DaWin generalize to tasks beyond classification benchmarks (e.g., NLP or reinforcement learning)?
2. Can the reliance on entropy as an expertise proxy limit DaWin's performance in certain scenarios, such as multi-modal data?
3. How does DaWin compare in terms of inference speed and energy consumption to AdaMerging and WiSE-FT?
4.What scenarios, if any, might lead to simpler methods outperforming DaWin, and how can this be systematically explored?

---

> ### Author Response · Authors · 2024-11-22
> **Response to the Reviewer yGMX [1/3]**
>
> Dear Reviewer yGMX,
> The authors are delighted that you find some advantages of our methods, including **training-free characteristics, novel use of entropy, state-of-the-art performance, and computational efficiency!**
> Your constructive comments made authors further deliberate on the limitation of the proposed method and raised a bunch of new insightful analyses that we are presenting through the responses below.
>
> ---
>
> > W1) Incremental novelty: While the method provides refinements over static interpolation, it shares similarities with existing methods (e.g., AdaMerging), which reduces its overall originality.
>
> `Answer:` DaWin has unique design motivation and advantages over existing methods. We would like to respectfully refute the novelty concern by providing three perspectives as below.
> * First, **DaWin aims to address dynamic interpolation scenarios in contrast to AdaMerging** which conducts a single interpolation and uses the model for all test samples. Meanwhile, based on the findings of our pilot study, we are motivated to build a dynamic interpolation method to achieve par better performance over static interpolation.
> * Second, unlike existing dynamic interpolation methods that require additional training and extra modules to compute per-sample interpolation coefficients, **DaWin is a training-free** method that can instantly be adopted given (pre-)trained models.
> * Lastly, **DaWin pursues efficiency by leveraging the Beta Mixture Modeling** approach which significantly reduces the amount of inference-time interpolation operations, and this design philosophy also discerns our method from previous dynamic interpolation methods.
>
> > W2/Q4) Over-reliance on entropy: The assumption that entropy reliably measures model expertise across all scenarios is not thoroughly validated, and alternative metrics could be explored further. What scenarios, if any, might lead to simpler methods outperforming DaWin, and how can this be systematically explored?
>
> `Answer1:` We already experimented with some alternatives (X-entropy loss based on different pseudo labels) in Figure 5 the experiment section of the manuscript. Those are representative replacements for entropy. The results showed that entropy achieves the best results overall.
> * We further conducted some experiments on other metrics for model expertise estimation: 1) maximum logit ratio, 2) exponentiated confidence ratio, and 3) exponentiated top1 and top2 confidence difference. The results are in the table below.
>
> | Method | ID Acc | OOD Acc Avg.  |
> |--------------------|-------------|-------------|
> | WiSE-FT        | 79.1         | 51.0        |
> | Max Logit Ratio        | 76.6         | 54.3        |
> | Confidence Diff. Ratio | 77.5         | 54.2        |
> | Confidence Ratio       | 77.5         | 54.2        |
> | Entropy Ratio (DaWin)  | 78.7         | 54.4        |
>
> * As shown in the table above, our entropy ratio-based design outperforms alternatvies in this setup.
>
> `Answer2:` We investigate a scenario when a simpler method can be preferred rather than DaWin.
> * As Reviewer yGMX noted, the entropy over individual test samples does not always reliably estimate each model's expertise if the uncertainty calibrations of individual models are bad.
> * We can investigate a potential failure mode of DaWin by controlling the level of uncertainty calibration of individual models, as the interpolation coefficients are computed from the output probability distribution through the entropy function.
> * In the table below, we analyze the effect of the individual model's calibration by dividing the pre-softmax outputs of each model with a scalar value.
>
> |   Method                              | ID Acc | OOD Acc  |
> |---------------------------------------|--------|----------|
> | WiSE-FT                                    | 79.1   | 51.0     |
> | DaWin                                      | 78.7   | 54.4     |
> | DaWin w/ bad calibration   | 76.6   | 54.0     |
>
> * As we can see, bad uncertainty calibration (manually adjusted toward under-confident prediction) of individual models makes DaWin fail to reliably estimate the per-sample coefficients and a simple method can be preferred in terms of ID and OOD performance trade-off.

---

> ### Author Response · Authors · 2024-11-22
> **Response to the Reviewer yGMX [2/3]**
>
> > W3/Q3) Trade-offs not fully explored: The paper does not sufficiently address the performance trade-offs between per-sample interpolation and simpler methods like weight averaging. How does DaWin compare in terms of inference speed and energy consumption to AdaMerging and WiSE-FT?
>
> `Answer1:` **We already provided the trade-off discussion as a separate paragraph in Section 6 of the manuscript including the inference speed, and we additionally provide results for other metrics here.**
> * We observed that DaWin can achieve outstanding classification accuracy that is comparable to or even better than training-intensive methods (AdaMerging) or fully sample-wise interpolation methods, while slightly increasing the wall-clock runtime compared to the hyperparameter tuning-intensive static interpolation methods.
> * Moreover, to further investigate the trade-offs between DaWin and a simpler method, we provide results on additional metrics (FLOPs and peak memory allocation) and variants of DaWin, in the table below.
>
> | Method  | Total Runtime (sec)| FLOPs per sample | Peak memory (MB) allocation per sample | OOD Avg. Acc|
> |:--------:|:--------------:|:--------------:|:--------------:|:--------------:|
> | WiSE-FT    | 1040       | 4.4139         | 454 |  51.0 |
> | DaWin |     1802     |   13.2452        | 1338 |  54.4 |
> | DaWin (parallel)  | 1341    | 8.8313     | 1338  |  54.4 |
>
> * Here, DaWin (parallel) denotes the setting where we parallelize the evaluations of individual models during the expertise estimation phase before conducting the interpolation. DaWin shows three times FLOPs than WiSE-FT, but the actual runtime is less than twice that of WiSE-FT because DaWin does not require intensive hyperparameter tuning, unlike WiSE-FT. Besides, the computational complexity can be further reduced by adopting the parallel inference pipeline for the model expertise estimation phase of DaWin.
>
> `Answer2:` Comparison for the energy consumption can be also discussed through the GPU hours (that would be multiplied with a GPU-specific power-consumption quantity), GPU memory allocation, and FLOPs (those are relevant to algorithmic efficiency in terms of required operation on CPUs and GPUs).
> * DaWin could not be better than the simpler method WiSE-FT in terms of energy consumption given increased runtime, FLOPs, and memory requirement.
> * However, not only for better energy efficiency, we believe that **ensuring robustness under distribution shifts (e.g., good OOD accuracy which can be remarkably improved by DaWin) is another crucial value that we should pursue for reliable AI as well [20].**

---

> ### Author Response · Authors · 2024-11-22
> **Response to the Reviewer yGMX [3/3]**
>
> > Q1) How well does DaWin generalize to tasks beyond classification benchmarks (e.g., NLP or reinforcement learning)?
>
> `Answer:` Although the scope of this work is limited to classification scenarios as noted in Section 2 of the manuscript, to investigate the applicability of DaWin beyond the classification setup, we additionally conduct experiments on the NLP domain.
> * By following **LoRA Hub [4] setup that adapts FLAN-T5 [10] with LoRA [3]**, among 196 trained LoRA weights, we randomly sampled 20 LoRA weights and conducted the dynamic interpolation per downstream tasks (Big Bench Hard [5]).
> * Different from the image classification setup, we use the negative log-likelihood (NLL) and perplexity (PPL) as model expertise measures that are tailored to the autoregressive language model evaluation.
> * Then, we leveraged few-shot samples (five input-output pairs used in the LoRA Hub method) to compute NLL and PPL. The average performance (exact matching) on 25 tasks from Big Bench Hard is in the table below.
>
> | Method    | Zero-shot | LoRA Hub | DaWin (PPL) | DaWin (NLL) |
> |-----------|-------|-------|-------|-------|
> | Avg. perf. | 29.18 | 36.21 | 36.55 | 36.61 |
>
> * We can see that DaWin using PPL and NLL both show better average performance compared to LoRA Hub which requires additional training per each downstream task.
> * Due to the lightweight nature of LoRA, DaWin only consumes 13% more memory allocation compared with the zero-shot pre-trained model inference.
>
> > Q2) Can the reliance on entropy as an expertise proxy limit DaWin's performance in certain scenarios, such as multi-modal data?
>
> `Answer:` As we mentioned in Section 2 of the manuscript, we confined the scope of this work to the multi-class classification problem and accordingly derived a reasonable proxy for the model expertise. If we consider multi-modal input-output data distribution with corresponding multi-modal models such as CoCa [11] and LLaVA [12], we may need to design new suitable expertise metrics tailored to the class of model rather than the output entropy which is suitable for the classification model. Investigating a new score tailored to those multi-modal models will be crucial for future work direction, and we provide the results on the NLP domain as an example for this extension, e.g., using negative log-likelihood or perplexity of language models' output as a proxy of model expertise in the above paragraph.

---

> > ### Author Response · Authors · 2024-11-26
> > **Invitation for discussion**
> >
> > Dear reviewer yGMX,
> >
> > Thanks again for taking your valuable time to give us constructive feedback.
> >
> > We have provided our rebuttals with additional empirical validation to address your concerns.
> >
> > The authors are sincerely wondering whether these responses are addressing your concerns.
> >
> > We would be happy to discuss our work with you, and looking forward to this.
> >
> > Best regards,
> >
> > The authors

---

### Official Review · Reviewer_uT7T · 2024-10-29

**Soundness:** 3
**Presentation:** 3
**Contribution:** 3
**Rating:** 6
**Confidence:** 3

**Summary:**

This paper describes a method of doing sample based parameter merging in order to improve performance on OOD test samples. The main idea is to estimate the entropy of each model and then to use this to perform the merging step. As this is quite slow, they also describe a faster mixture-modelling technique. They first demonstrate with 'GT' data that if you had an oracle for the entropy you would do well. They then show that this oracle / estimated entropy is correlated and finally that they achieve superior results over similar methods.

**Strengths:**

The paper is quite interesting and presents their main idea clearly. I summarise the strengths below:
1. Clarity: the paper is clear and gets across the main message (can we use an estimate of entropy at the sample level in order to improve OOD performance).  I like the 'GT' set up as it clearly shows the potential improvements of such an approach and gets across the intuition of why this is useful.

2. Experiments: they have a range of results backing up their claims and showing that their method improves over the baselines. They further generalise their approach to other settings (e.g. dynamic classifier selection or ensemble selection)

3. The consider compute and performance and show how they are trading off compute (though maybe not as badly as others) for improved performance.

**Weaknesses:**

1. Clarity: There are a few parts of the paper that are lacking details or are confusing:
1a. in the intro a foundational model != ImageNet. People refer to foundational models primarily for large scale models trained on vast amounts of data (GPT,Gemini,Flamingo)
1b. It is unclear how precisely H(-) is computed. I appreciate this is probably obvious to the authors but it would be helpful to briefly mention.
1c. I am unconvinced by their linear mode connectivity explanation. My understanding is the point of the linear mode connectivity is that *along that line between model weights*, the error is bounded to some height. But this is *not* what's being checked -- it's being done for a single interpolated example.

**Questions:**

Please refer to questions in weaknesses but here are additional questions that are not weaknesses but I'm curious about:

1. Did they see any trends in where they obtained similar interpolation coefficients for samples? Maybe certain clusters of samples are always better explained by a given model or not. Are there conclusions one could draw about the test set from this ?

---

> ### Author Response · Authors · 2024-11-22
> **Response to the Reviewer uT7T [1/2]**
>
> Dear Reviewer uT7T,
> The authors appreciate your productive feedback (clarity suggestions and a valuable question), which helped the authors significantly refine the paper overall, and we are thankful for acknowledging the **strengths of this paper: clear presentation, including motivating example (GT scenario), solid experiments, and trade-off analysis.** We provide the responses to your comments as below.
>
> ---
>
> > Clarity: There are a few parts of the paper that are lacking details or are confusing: 1a. in the intro a foundational model != ImageNet. People refer to foundational models primarily for large scale models trained on vast amounts of data (GPT,Gemini,Flamingo)
>
> `Answer:` In the introduction section of our manuscript, we do not use the term '_foundation model_' to mean the ImageNet pre-trained model. The ImageNet and ImageNet-A are mentioned to illustrate the standard setup of robust fine-tuning scenario [6,7,8] where we fine-tuned a foundation model (CLIP). The CLIP-like vision-language models have been regarded as a representative class of foundation models [9,10,11] besides with large language models.
>
> > Clarity: 1b. It is unclear how precisely H(-) is computed. I appreciate this is probably obvious to the authors but it would be helpful to briefly mention.
>
> `Answer:` Does reviewer uT7T wonder about how precise the estimation of model entropy is to the entropy over true conditional data distribution $P(Y|X)$?
> * Unfortunately, we can not compute the true entropy because we do not have the true data-generating process on the benchmark datasets, so can not measure the quality of entropy estimation.
> * However, as we adopted entropy as a proxy of X-entropy, we can alternatively compute the correlation between entropy and X-entropy. The Pearson correlation coefficients between entropy and X-entropy are provided below.
> *
> | Model  | ImageNet | ImageNetV2 | ImageNetR | ImageNetA | ImageNetSketch | ObjectNet|
> |:------:|:--------:|:----------:|:---------:|:---------:|:--------------:|:--------------:|
> | Model0 | 0.515    | 0.513      | 0.664     | 0.267     | 0.579          | 0.483 |
> | Model1 | 0.435    | 0.417      | 0.589     | 0.064     | 0.4            | 0.327 |
>
> * As we can see, each model's entropy largely shows moderate correlations [18] with the X-entropy computed with ground truth labels.
>
> > Clarity: 1c. I am unconvinced by their linear mode connectivity explanation. My understanding is the point of the linear mode connectivity is that along that line between model weights, the error is bounded to some height. But this is not what's being checked -- it's being done for a single interpolated example.
>
> `Answer:` We derived a new hypothesis analogy to linear mode connectivity (LMC) which is adapted to entropy setup. **By reflecting the Reviwer uT7T's concern, we conduct an experiment to empirically verify this hypothesis**.
> * The sample-wise entropy valley hypothesis claims that 1) there exists a global coefficient $\lambda$ that the average entropy of the interpolated model is smaller than the minimum of two models' average entropy, 2) there also exists a function $\lambda(x)$ that determine per-sample coefficients such that average entropy from the interpolated models becomes much smaller than that of a static interpolation.
> * In Figure H of the Appendix in the revised manuscript, we present individual models' entropy, static interpolation models' entropy, and dynamic interpolation models' entropy. The results aligned with our two hypotheses and therefore empirically support our statements (**Please refer to Figure H of Appendix**).

---

> ### Author Response · Authors · 2024-11-22
> **Response to the Reviewer uT7T [2/2]**
>
> > Q) Did they see any trends in where they obtained similar interpolation coefficients for samples? Maybe certain clusters of samples are always better explained by a given model or not. Are there conclusions one could draw about the test set from this ?
>
> `Answer:` Yes. We observed that in the robust fine-tuning (ImageNet) setup, the estimated coefficients are diverse within a dataset, whereas in the multi-task learning (Specialized visual recognition) setup, the estimated coefficients are mostly similar across samples within a dataset.
> * **Explanation**) If there exist some dominant expert models for a test set, and the train sets of those dominant experts are remarkably similar to the test set compared to non-dominant models' ones, the estimated interpolation coefficients will be similar across test samples. Meanwhile, the opposite case (coefficients are diverse across samples) implies that the degree distributional similarities between the test set and train set of each model are roughly equal.
> * As an extreme example, let's assume the train datasets of individual models are highly specialized to some domains that have narrow distributions, and the test set is also one of highly specialized task-specific data. Then, **DaWin will induce similar coefficients over entire test samples that are skewed toward a model whose train distribution is the most similar to the test distribution.**
> * To systematically investigate this, we simulate some scenarios in which models used to merge have different levels of relevancy to the target test dataset.
>
> | Scenario               | Train datasets    | Test dataset  |  STD   |
> |------------------------|-------------------|---------------|--------|
> | skewed relevancy   | EuroSAT, MNIST    | SVHN          | 0.098  |
> | skewed relevancy   | EuroSAT, SVHN     | MNIST         | 0.083  |
> | skewed relevancy   | SUN397, MNIST     | SVHN          | 0.102  |
> | skewed relevancy   | SUN397, SVHN      | MNIST         | 0.075  |
> | balanced relevancy     | EuroSAT, RESISC45 | SUN397        | 0.162  |
> | balanced relevancy     | EuroSAT, SUN397   | RESISC45      | 0.192  |
> | balanced relevancy     | RESISC45, SUN397  | EuroSAT       | 0.135  |
> | balanced (ir)relevancy | EuroSAT, MNIST    | DTD           | 0.173  |
> | balanced (ir)relevancy | EuroSAT, SVHN     | DTD           | 0.151  |
> | balanced (ir)relevancy | SUN397, MNIST     | DTD           | 0.171  |
> | balanced (ir)relevancy | SUN397, SVHN      | DTD           | 0.130  |
>
> * The above table presents the standard deviation (STD) of DaWin's per-sample estimated coefficients for each scenario.
> * We observed that, in the skewed relevancy setups where one of the models has remarkably better expertise compared to another one, the STDs are much smaller than that derived from balanced relevancy setups where two models are trained on similarly relevant datasets or similarly irrelevant ones regarding test distribution.
> * We conclude that the similarity among obtained coefficients depends on the relation between the test dataset and the datasets where individual models are trained.

---

> ### Author Response · Authors · 2024-11-26
> **Invitation for discussion**
>
> Dear reviewer uT7T,
>
> The authors want to express their appreciation for your productive feedback and for taking the time to review this work.
> We have presented our rebuttals with additional experiments to address your concerns.
> The authors are wondering whether these are addressing your concerns.
> We are sincerely looking forward to further discussing our work with you who has remarkably contributed to improving the quality of the manuscript so far.
>
> Best regards,
>
> The authors

---

### Official Review · Reviewer_Rb5n · 2024-10-30

**Soundness:** 3
**Presentation:** 3
**Contribution:** 2
**Rating:** 6
**Confidence:** 4

**Summary:**

This paper proposes a dynamic weight interpolation method DaWin. It merges pre-trained and fine-tuned models based on per-sample entropy without training. It further uses Beta mixture models to cluster interpolation coefficients to keep computation efficient. DaWin achieves considerable performance, offering a scalable solution for adaptable model merging.

**Strengths:**

1. DaWin requires no additional training, making it highly practical for real-world deployment.
2. The usage of Beta mixture models ensures adaptation scalability, reducing the inference-time overhead of dynamic interpolation.
3. DaWin shows its performance on both robust fine-tuning and multi-task learning problems.
4. DaWin does not require extensive hyperparameter tuning.

**Weaknesses:**

1. The model utilizes Beta Mixture Models for efficiency, but only complexity and wall-clock time results are provided. There are no reported MACs or FLOPs per sample, limiting insights into computational costs.

2. Since the model performs per-sample interpolation, it requires three inferences per sample, increasing computational demands.

3. The dynamic interpolation adjustment can be viewed as an incremental improvement over existing interpolation methods, rather than a novel breakthrough. The level of novelty remains unclear.

4. While the paper presents a comprehensive experimental study, the explanations for the results are shallow. There is limited discussion on why DaWin performs well or poorly on specific datasets or distributions (e.g., SUN397 and Cars). Additionally, relationships among datasets are critical but remain under-explored.

5. The entropy-based expert assumption is built on simple object concepts, implying that the datasets are sufficiently learnable through fine-tuning a CLIP model, where entropy should ideally align with correct predictions. However, in real applications, entropy can be misleading, resulting in overconfident but incorrect predictions. The model design does not address how to correct these wrong predictions. While the TrueTrue/TrueFalse breakdown in Figure 2 is insightful, it is not directly leveraged in the experiments, which limits the depth of the analysis.

**Questions:**

1. From my point of view, the dynamic weighting strategy per sample is another form of confidence-based selection, where the preference is determined solely by model confidence. Given this, what if we fuse the outputs instead of the model weights? Do the tasks (e.g., ImageNet - ImageNet-A) reveal any trends in the weighting of these models?

2. How can we determine whether the performance is correlated with the **distribution distances** between datasets?

3. Could the proposed method result in negative weight mixtures for some datasets? If so, why?

4. Is it possible that some checkpoints could **decrease** the final performance on specific datasets? Furthermore, do the semantic distances and relationships between classes from different datasets (and their corresponding checkpoints) affect the performance of one another?

5. For the evaluation of wall-clock time, is it tested per dataset? or per sample?

6. Please also review the drawbacks section.

---

> ### Author Response · Authors · 2024-11-22
> **Response to the Reviewer Rb5n [1/5]**
>
> Dear Reviewer Rb5n,
> The authors sincerely appreciate your valuable comments that extensively cover the in-depth discussion on the technical details, algorithmic limitations, further analysis, and explanation of the results!! We strongly believe that all of this constructive feedback greatly contributes to improving the quality of this work. We also thank for the acknowledgement from you on the **strengths of this paper: training-free, Beta mixture based efficiency-seeking design, outstanding performance of diverse experiments, and hyperparameter tuning-free nature!** We address your concerns and questions in detail through five threads below.
>
> ---
>
> > W1/W2/Q5) There are no reported MACs or FLOPs per sample, limiting insights into computational costs. Since the model performs per-sample interpolation, it requires three inferences per sample, increasing computational demands. For the evaluation of wall-clock time, is it tested per dataset? or per sample?
>
> `Answer:`
> * We additionally measure **FLOPs per sample** for each method (CLIP ViT-B/32 robust fine-tuning setup).
> * The **wall-clock time** in the manuscript and the table below is **aggregated for all evaluation datasets**.
> * Meanwhile, in terms of **computational demands**, we can further lessen the computational complexity during inference _by performing expertise estimations (forward evaluations) of individual models in parallel given multiple GPUs_ (two GPUs for this experiment). Note that this parallelism is orthogonal to the commonly used distributed data-parallel (DDP) inference method and is only applicable to DaWin. The results of applying those parallelizations are shown in the table below.
>
> | Method  | Total wall-clock time (sec)| FLOPs per sample | OOD Avg. Acc|
> |:--------:|:--------------:|:--------------:|:--------------:|
> | WiSE-FT    | 1040       | 4.4139          |   51.0 |
> | WiSE-FT w/ DDP    | 825       | 4.4139          |   51.0 |
> | DaWin |     1802     |   13.2452         |  54.4 |
> | DaWin (parallel)  | 1341    | 8.8313      |  54.4 |
> | DaWin (parallel) w/ DDP | 1064  | 8.8313      |  54.4 |
>
> * Although the FLOPs of DaWin are three times larger than WiSE-FT they can be reduced by adopting the parallelization technique.
> * It is worth noting that our algorithmic parallelism induces 134% faster inference compared to DDP which makes an improvement of 126% than that of vanilla inference.
> * Moreover, in terms of total wall-clock time which measures the required time for all evaluations (including hyperparameter tuning of WiSE-FT), it does not increase by three or two times, unlike FLOPs, because DaWin does not require hyperparameter tuning.

---

> ### Author Response · Authors · 2024-11-22
> **Response to the Reviewer Rb5n [2/5]**
>
> > W3) The dynamic interpolation adjustment can be viewed as an incremental improvement over existing interpolation methods, rather than a novel breakthrough.
>
> `Answer1:` We respectfully refute novelty issues from two perspectives: the design of the method and actual performance improvement.
> * In terms of **design of method**
>   * It is worth noting that DaWin is a pioneering **training-free dynamic interpolation** method which is greatly underexplored so far.
>   * As mentioned in the third paragraph of the introduction of the manuscript, **existing methods have required additional training** and extra modules to dynamically determine the interpolation coefficient. However, **DaWin does not require any training** while also addressing the computation complexity issue.
>   * To summarize, we provided a scalable solution that addresses some intrinsic challenges of dynamic interpolation: (a) training-free: computing proper per-sample interpolation coefficients without training by just relying on the forward evaluation of each model; (b) efficient dynamic interpolation: reduced computational complexity through Beta Mixture.
> * In terms of **performance improvement**
>   * We would like to emphasize that WiSE-FT's performance on the ID-OOD performance in Figure 4 of the manuscript can not be achievable given the fact that we cannot select the interpolation coefficient based on OOD performance because we do not have test set labels in advance. Therefore, we get the performances of WiSE-FT by tuning its interpolation parameter on ID validation set in practice.
>   * Given that fact, in terms of average OOD accuracy of CLIP VIT-B/32, B/16, L/14 experiments, our method achieves significantly better performance compared with existing methods including WiSE-FT and the state-of-the-art method, Model Stock [18].
>
> `Answer2:` **Determining appropriate per-sample interpolation coefficients is a challenging non-trivial problem, which we address in this work for the first time.**
> * In the table below, we provide some dynamic interpolation coefficient designs from two categories (data-independent and data-dependent).
>
> | Category             | Coefficient per sample | ID Acc | OOD Acc Avg.  |
> |----------------------|------------------------|--------------|-------------|
> | No interpolation (FT)| -                      | 78.4         | 47.1        |
> | Static interpolation | 0.8 (WiSE-FT)          | 79.1         | 51.0        |
> | Data-independent     | Uniform$(0,1)$         | 75.9         | 52.9        |
> | Data-independent     | Normal$(0.5,0.2^{2})$  | 77.5         | 54.1        |
> | Data-independent     | Normal$(0.8,0.1^{2})$  | 79.1         | 51.1        |
> | Data-dependent       | Max Logit Ratio        | 76.6         | 54.3        |
> | Data-dependent       | Confidence Diff. Ratio | 77.5         | 54.2        |
> | Data-dependent       | Confidence Ratio       | 77.5         | 54.2        |
> | Data-dependent       | Entropy Ratio (DaWin)  | 78.7         | 54.4        |
>
> * The results show that DaWin's entropy ratio-based coefficients greatly outperform other reasonable alternatives in terms of ID and OOD performance trade-offs. That is, some alternative methods can achieve good OOD performance but they remarkably hurt ID Acc compared to DaWin.
> * This demonstrates the effectiveness of our unique design choice which is well-grounded with our unique motivation from the pilot study in Section 3 of our manuscript.

---

> ### Author Response · Authors · 2024-11-22
> **Response to the Reviewer Rb5n [3/5]**
>
> > W4) There is limited discussion on why DaWin performs well or poorly on specific datasets or distributions (e.g., SUN397 and Cars). Additionally, relationships among datasets are critical but remain under-explored.
>
> `Answer:` We provide a **hypothesis and supporting empirical evidence on the performance variation of DaWin depending on the characteristics of datasets**, and further provide a **suggestion** for successful usage of DaWin. To summarize, **DaWin's success may depend on the difference between a number of classes of train datasets.**
> * We appreciate Reviewer Rb5n's constructive suggestions on further analysis. By reflecting the feedback, we compute the distance between datasets with mean maximum discrepancy (MMD) and Frechet inception distance (FID) to analyze the performance of DaWin in terms of relationships between datasets. Please refer the Appendix Figure G of our revised manuscript to see this.
> * To deepen our understanding of dataset-specific performance variation of DaWin, we analyzed the proportion of samples that the true expert (e.g., model fine-tuned on DTD train split for DTD evaluation) produced the smallest entropy so that get the largest weight for merging among individual models. **This is directly matched with the entropy-dominancy assumption that we assume in Lemma 6.1. as DaWin's working condition**.
>
> | Dataset  | most relevant model (true expert) | second-most relevant model  | sum of both  |
> |:--------:|:--------------------:|:--------------:|:-----------------------:|
> | **SUN397**   | 0.8153               | 0.0393         | **0.8546**                  |
> | **Cars**     | 0.8556               | 0.0126         | **0.8682**                  |
> | RESISC45 | 0.9610                | 0.0219         | 0.9829                  |
> | EuroSAT  | 0.9967               | 0.0033         | 1.0000                  |
> | SVHN     | 0.6252               | 0.3739         | 0.9991                  |
> | GTSRB    | 0.9992               | 0.0002         | 0.9994                  |
> | MNIST    | 0.9963               | 0.0035         | 0.9998                  |
> | DTD      | 0.9426               | 0.0186         | 0.9612                  |
>
> * **Observation**: In the table above, we can see that **SUN397 and Cars** datasets, where DaWin's performances are farthest from the ideal individual models' performances, show the lowest proportion of samples that meet entropy-dominancy assumption thereby incorrectly weighing each model.
>     * Although SVHN shows the lowest proportion in terms of 'most relevant model', SVHN dataset shows strong similarity with MNIST dataset in terms of MMD and FID, and the summation of the entropy-dominancy samples' proportions in those two datasets approach 100% likewise other datasets.
>     * However, in SUN397 and Cars datasets, roughly 85% of samples only meet the entropy-dominancy assumption and 15% of samples fail to correctly give the largest merging weight for the true expert.
>     * Therefore, we speculate that DaWin's success or failure depends on the proportion of samples meeting entropy-dominancy assumption.
> * **Possible explanation**: We conjecture that the difference between **number of classes** of downstream tasks related to the proportion of entropy-dominancy samples in that dataset.
>     * Intuitively, as the dimensionality of the output probability distribution becomes larger, the entropy over that distribution becomes larger in general because there are more possible states and outcomes.
>     * Therefore, models trained on such datasets (SUN397 and Cars have 397 and 196 classes, respectively) become less confident about their prediction (higher entropy), so the proportions of entropy-dominancy samples become smaller compared to other datasets confined to a small number of classes (10 ~ 47 for remaining six datasets).
> * **Suggestion**: As long as we adopt the entropy as an expertise metric for individual models, one should be cautious to use DaWin in the scenario where individual models are trained on datasets those the numbers of classes are very different. We recommend using DaWin in the setting where the numbers of classes are well-matched to each trained model, and if not, other quantities such as pseudo label-based X-entropy loss can be adopted as an alternative model expertise metric.
> * The relationships among datasets will be further explored in the following paragraph.

---

> ### Author Response · Authors · 2024-11-22
> **Response to the Reviewer Rb5n [4/5]**
>
> > Q2/Q4) How can we determine whether the performance is correlated with the distribution distances between datasets? Is it possible that some checkpoints could decrease the final performance on specific datasets? Furthermore, do the semantic distances and relationships between classes from different datasets (and their corresponding checkpoints) affect the performance of one another?
>
> `Answer1:` We can systematically analyze the **relation between distributional distances between datasets and the performance of merged method** by constructing a model pool for merging based on the distance between train datasets (where the candidate models are trained) and test datasets. Please refer the Appendix Figure G of our revised manuscript to see the the heatmap of pairwise dataset distances.
> * We simulate two scenarios: 1) all models are relevant to solve the downstream task, and 2) some models are relevant but others are not to solve the downstream task. The results are in the table below.
>
> | Scenario                | Train                         | Eval | Task Arithmetic | AdaMerging++ | DaWin  |
> |:-----------------------:|:-------------------------------------:|:-------------:|:---------------:|:------------:|:------:|
> | Experts only | SVHN, MNIST        | SVHN      | 87.5            | **94.1**         | 90.5   |
> | Mixed | SVHN, MNIST, EuroSAT            | SVHN      | 84.4            | 93.7         | **93.8**   |
> | Mixed | SVHN, MNIST, EuroSAT, RESISC45 | SVHN      | 81.5            | 93.4         | **94.8**   |
> | Experts only | EuroSAT, RESISC45     | EuroSAT   | 96.4            | 98.1         | **98.6**   |
> | Mixed | SVHN, MNIST, EuroSAT            | EuroSAT   | 86.9            | 97.7         | **99.6**   |
> | Mixed | SVHN, MNIST, EuroSAT, RESISC45 | EuroSAT  | 89.6            | 97.7         | **99.7**   |
>
> * Although the **baseline methods** perform well when we have expert models trained on datasets that are relevant to the evaluation dataset, they **suffer from performance degeneration as the non-expert models are included** in the merging pool. This indicates that there are some interferences between models (and train corresponding datasets) that hurt post-merging performance if we inappropriately determine the interpolation coefficient.
> * Meanwhile, **DaWin takes benefits from even non-expert models by leveraging entropy ratio-based weighting** strategy, and the performance is improved as more models participate in merging.
>
> `Answer2:` We further analyze whether we can take any insight on the impact **relationship between classes from different datasets** on DaWin.
> * Characterizing semantic distances between classes from different datasets, and relating them with post-interpolation performance is a non-trivial problem. Given that **DaWin relies on the output probability distribution of each model, we measure and analyze the pairwise kullback leibler divergence (KLD) over the averaged per-class output probability** distributions from each fine-tuned model.
> * The mean of pairwise KLD over per-class output distributions measures how different the output probability distribution is between each class, and is related to class-wise distinctiveness and separability in the output space.
> * We hypothesize that as the mean pairwise KLD of the dataset is lower, the strength of the model (trained on that dataset) expertise becomes weaker, and does not much contribute during merging when evaluating the training dataset itself or other datasets. The results are in the table below.
>
> | Dataset  | Pairwise KLD mean | Mean coef. for train dataset  | Mean coef. for other datasets  | Sum of both |
> |:--------:|:-----------------:|:---------------:|:---------------:|:---------------:|
> | SUN397   | 0.021            | 0.3000       | 0.0779 | 0.3779 |
> | Cars     | 0.054           | 0.3012         | 0.0682 | 0.3694 |
> | RESISC45 | 0.122            | 0.3401         | 0.0854| 0.4255 |
> | EuroSAT  | 0.363           | 0.3445        | 0.0745 | 0.4190 |
> | SVHN     | 0.284           | 0.3715        | 0.0906 | 0.4621 |
> | GTSRB    | 0.156        | 0.5054          | 0.0884 | 0.5938 |
> | MNIST    | 0.559             | 0.3716      | 0.1551 | 0.5267 |
> | DTD      | 0.086            | 0.4024          | 0.0856 | 0.4880 |
>
> * We see that the mean of pairwise KLD of each dataset somewhat negatively correlated with the magnitude of the coefficient of each corresponding model on the training dataset and other datasets. For example, the model trained on Cars dataset shows the lowest contribution during merging when evaluating on Cars or other datasets.
> * We speculate that class-wise relationship and separability over a dataset can be a measure of the strength of expertise of the model trained on that dataset, which is crucial to the success of DaWin.

---

> ### Author Response · Authors · 2024-11-22
> **Response to the Reviewer Rb5n [5/5]**
>
> > Q3) Could the proposed method result in negative weight mixtures for some datasets? If so, why?
>
> `Answer:` We use the softmax function to ensure that DaWin's coefficient vectors have elements ranging from 0 to 1. Although DaWin can generate zero-approaching values for some experts that produce higher entropy than other models, it cannot have negative values.
>
> > W5) In real applications, entropy can be misleading, resulting in overconfident but incorrect predictions. The model design does not address how to correct these wrong predictions.
>
> `Answer1:` As Reviewer Rb5n noted, the entropy over individual samples does not always align with the model's predictiveness because there are inevitably some corner cases in which a model produces over-confident prediction while it is incorrect. However, **we indeed consider some techniques to address those overconfidence and incorrect individual predictions in our design of the method.**
> 1. We adopted temperature scaling [15], which is mentioned in the implementation detail section of the Appendix.
> 2. Moreover, as we noted in L255 of Section 3.2., we leverage domain-specific offset adjustment to refine the unstable sample-wise entropy ratios of some corner cases. This offset adjustment consistently improved the performance in all cases (Please refer to Table A of the Appendix for the ablation study).
>
> We revised the manuscript to inform this more remarkable manner at the end of Sec 3.2.
>
> `Answer2:` The rationale behind using entropy is laid on the fact that it can well approximate X-entropy (the desired expertise metric). In the case where entropy is not a suitable model expertise metric such as the autoregressive language model [12,19], **alternative quantities such as negative log-likelihood (NLL) or perplexity (PPL) can be adopted as an expertise metric**.
>
> * To simulate such a setting, we conduct an experiment on an autoregressive language model.
> * By following LoRA Hub [4] setup that aims to fine-tune FLAN-T5 [10] via LoRA [3], we randomly sampled 20 public LoRA weights and performed the dynamic interpolation per downstream tasks (Big Bench Hard [5]).
> * We adopt the negative log-likelihood (NLL) and perplexity (PPL) as model expertise metrics and leverage few-shot samples (five input-output pairs used in LoRA Hub) to compute NLL and PPL. The average matching accuracy on 25 tasks is in the table below.
>
> | Method    | Zero-shot | LoRA Hub | DaWin (PPL) | DaWin (NLL) |
> |-----------|-------|-------|-------|-------|
> | Avg. perf. | 29.18 | 36.21 | 36.55 | 36.61 |
>
> * We see that DaWin using PPL and NLL both show better performance compared to LoRA Hub which requires additional training per each downstream task.
>
>
> > Q1) the dynamic weighting strategy per sample is another form of confidence-based selection. Given this, what if we fuse the outputs instead of the model weights? Do the tasks (e.g., ImageNet - ImageNet-A) reveal any trends in the weighting of these models?
>
> `Answer:` Yes. The entropy ratio-based weighting strategy can be leveraged to fuse the model output as well as model parameter, and **we already conducted the applications of DaWin for output ensemble** setup in Table 5 of our manuscript. We provide a full table for the dynamic output ensemble (DOE) method.
>
> | Method   | ImageNet | ImageNetV2 | ImageNetR | ImageNetA | ImageNetSketch | ObjectNet  |
> |----------|----------|------------|-----------|-----------|----------------|------------|
> | ZS       | 63.35    | 55.88      | 69.26     | 31.44     | 42.32          | 43.49      |
> | FT       | 78.35    | 67.22      | 59.28     | 24.66     | 42.22          | 42.03      |
> | WISE-FT  | 79.14    | 68.37      | 65.40     | 29.39     | 46.02          | 45.91      |
> | DOE      | 78.59    | 67.92      | 71.53     | 30.99     | 47.41          | 46.49      |
> | DaWin    | 78.70    | 68.41      | 72.12     | 34.41     | 48.34          | 48.49      |
>
> * Similar to DaWin (weight interpolation), DOE shows relatively weak performance on ID (ImageNet) and easy OOD (ImageNetV2 which is generated by reproducing the ImageNet data collection pipeline a decade later) while outperforming on more semantically different OOD datasets compared with static weight interpolation baseline (WiSE-FT).

---

### Official Review · Reviewer_sGAj · 2024-11-03

**Soundness:** 3
**Presentation:** 4
**Contribution:** 3
**Rating:** 8
**Confidence:** 3

**Summary:**

A new method for robust fine-tuning, called "dynamic weight interpolation" (DaWin) is proposed. It is directly related to the well known model averaging method WiSE-FT. However, while WiSE-FT averages model parameters, using a fixed coefficient $\lambda$ that does not depend on the input data, DaWin proposes to make the coefficient data dependent, i.e., to use $\lambda(x)$. The optimal $\lambda(x)$ is computed, using a heuristic that is based on the entropy of the model output. DaWin is tested against many existing robust fine-tuning methods, yielding slightly better OOD performance compared to WiSE-FT.

**Strengths:**

In general, the paper written in a clear and understandable way. It provides novel experimental evidence, that averaging the weights of pre-trained and fine-tuned models to keep OOD performance high might not be fine grained enough. The proposed heuristic to select sample based averaging coefficients intuitively makes sense and is backed up by well designed experiments.
The performance and computational complexity of DaWin is compared to a sensible selection of existing methods.

**Weaknesses:**

I see two problems:

1) In section 2.2, the a pilot study is provided to show the necessity of sample-wise averaging coefficients $\lambda(x)$ for good out of distribution performance after fine-tuning. As I understood, the goal is to find the best possible OOD performance that we can achieve with an oracle $\lambda^*(x)$ for the sample-wise coefficients that actually knows the true labels for a given $x$.

The paper states, that such an oracle would be $\lambda^*(x) = p_{\theta_1}(y|x) / (p_{\theta_0}(y|x) + p_{\theta_1}(y|x))$, i.e., the probability whether data $(x,y)$ comes from the distribution $p_{\theta_1}(y|x)$. However, while this would make sense if we merge the output of the models (analogous to classifier selection), I cannot imagine that it is the seeked best oracle for $\lambda^*(x)$ if we perform weight averaging. To obtain this oracle, we would have to test for each $x$, if there exists a $\lambda^*(x)$, such that the model with parameters $\theta_{\lambda^*}$ predicts the correct $y$.

In the following sections, the paper makes some approximations to compute $\lambda^*(x)$ based on the entropy of the model output, which does not require the knowledge of $y$. The reader is lured into the impression, that the only performance gap to the oracle is caused by these approximations. However, I think this is not true, since the oracle itself might not be optimal. However, this issue is about presentation and could be fixed.

2) I think performance gains compared to WiSE-FT and other methods are often minimal. Although DaWin does not require fine-tuning, there is still a considerable memory overhead. To be able to perform per-sample averaging, we need to store both sets of weights. For large models, this might be a severe problem that is not worth the effort.

**Questions:**

* Could you please comment on the memory requirement of DaWin and how it compares to other methods, for both with and without mixture modeling?
* I am also not so sure about your provided costs for inference in Table 2. Could you point me where $H$ is defined and elaborate a bit the provided terms? Am I correct, that DaWin (without mixture modeling) requires 3 model evaluations (2 to compute the entropy for the individual models for a given $x$ and 1 for the averaged model)?
* In the experiments, DaWin sacrifices ID performance for better OOD performance compared to WiSE-FT. Could you elaborate in which application scenario I would want to choose DaWin over WiSE-FT. I would think in practice good ID performance still wins over good OOD performance if you cannot have both.

---

> ### Author Response · Authors · 2024-11-22
> **# Response to the Reviewer sGAj [1/3]**
>
> Dear Reviewer sGAj,
> The authors greatly appreciate your invaluable comment, such as the optimality of the oracle coefficients, from a deep understanding of our work. Your productive comments and questions help the authors to further analyze and understand the strengths and weaknesses of the proposed method and the paper itself. We also are thankful for your awareness of **the strengths of this work: clear and understandable writing, supporting empirical results, and addressing the computational complexity issue!** We provide our responses to your query in the three threads as below.
>
> ---
>
> > W1) ... while this would make sense if we merge the output of the models (analogous to classifier selection), I cannot imagine that it is the seeked best oracle for if we perform weight averaging. This issue is about presentation and could be fixed.
>
> `Answer:` As the Reviewer sGAj pointed out, $\lambda^{*}(x)$ is **not guaranteed to be optimal for the weight interpolation scenario.** We did _not_ use the term '_oracle_' to indicate '_optimal_', and rather use it to denote the setting where we use the **ground truth test labels**. We revised the manuscript to avoid potential misunderstanding from readers (Please See the footnote of Sec 2.2 and Table C in the Appendix of our revised PDF).
>
> * The rigorous 'optimality' of interpolation coefficients can be only discussed by manually searching those coefficients that induce the interpolated models to produce minimum loss per sample.
> * We conduct an empirical validation investigating how close our X-entropy ratio-based oracle coefficients are to the optimal coefficients that minimize the loss. Specifically, we computed optimal coefficients that minimize the X-entropy of the interpolated model by grid-searching over the test set. Then, we compared those optimal coefficients with our X-entropy ratio-based coefficients to measure the differences between them.
> * In the table below, we compared the mean squared error (MSE) between optimal coefficients and 1) Uniform(0,1) random variables, 2) WiSE-FT's single coefficient (0.8), and 3) ours.
>
> |      | ImageNet | ImageNetV2 | ImageNetR | ImageNetA | ImageNetSketch | ObjectNet  |
> |:-----------------:|:--------:|:----------:|:---------:|:---------:|:--------------:|:----------:|
> | STD of optimal coef.   | 0.2827   | 0.3068     | 0.3237    | 0.3287    | 0.3413          | 0.3438     |
> | Uniform$(0,1)$    | 0.1797   | 0.1823     | 0.1869    | 0.1976    | 0.198          | 0.2013     |
> | WiSE-FT constant | 0.1084   | **0.1390**      | 0.2081    | 0.2427    | 0.2036         | 0.2131     |
> | X-entropy ratio   | **0.0382**   | 0.1559     | **0.0479**    | **0.0364**    | **0.040**         | **0.044**      |
>
> * As we can see, compared with baseline (uniform randon varibles, a tuned WiSE-FT constant), our X-entropy ratio-based per-sample coefficients shows less MSE overall.
> * Given that the standard deviation of oracle coefficients are about from 0.28 to 0.34, the closeness between optimal and X-entropy ratio-based coefficients is remarkable.
>
> > W2) minor accuracy improvement given memory overhead
>
> `Answer:` Note that we could not access the true label during inference time. Accordingly, we could not achieve the best trade-off points of WiSE-FT as like in Figure 4., and the possible alternative is just tuning the interpolation coefficient of WiSE-FT on the ID validation set we have. The results from this strategy are presented in Tables 2, 3, 4. **In these cases, DaWin significantly outperforms WiSE-FT in terms of OOD accuracy.** The concern related to the memory overhead will be addressed in the response below.

---

> ### Author Response · Authors · 2024-11-22
> **# Response to the Reviewer sGAj [2/3]**
>
> > Q1) Could you please comment on the memory requirement of DaWin and how it compares to other methods, for both with and without mixture modeling?
>
> `Answer 1:` Here, we provide the memory requirement of DaWin in terms of storage and inference time peak memory usage. The mixture modeling does not affect the storage and peak memory usage and only affects the number of interpolation operations during inference, so DaWin in the table below denotes whether mixture modeling is applied or not.
>
> * Compared to static interpolation methods such as WiSE-FT, DaWin requires that all individual models should be saved in storage. Thus, the memory storage requirement of DaWin is twice timed to static merging in the robust fine-tuning setup.
> * Meanwhile, the inference time peak memory allocation by DaWin is roughly three times that of WiSE-FT because it requires loading two individuals and the interpolated model on memory. The results are summarized in the table below.
>
> | Method  | Storage requirement (MB) | Peak memory (MB) per sample | OOD Avg. Acc|
> |:--------:|:--------------:|:--------------:|:--------------:|
> | WiSE-FT |     435      |   454     | 51.0 |
> | DaWin |     870    | 1338 | 54.4 |
>
>
> * We acknowledge the memory overhead of DaWin as a weakness that should be mitigated in future work, but we believe that remarkably improved accuracy on OOD datasets overwhelms that weakness given that robustness under distribution shifts is increasingly crucial for reliable AI [20].
>
> `Answer 2:` If our application scenario is a parameter-efficient fine-tuning (PEFT) regime [1,2,3] rather than the full model fine-tuning scheme, this memory overhead may vanish.
> * To investigate whether DaWin can be effective in such setting, we conduct an experiment on PEFT of a language model.
> * By following LoRA Hub [4] setup that adapts FLAN-T5 [10] via LoRA [3], we randomly sampled 20 LoRA weights and conducted the dynamic interpolation per downstream tasks (Big Bench Hard [5]). Here we use the negative log-likelihood (NLL) and perplexity (PPL) as model expertise measure and leverage few-shot samples (five input-output pairs used in LoRA Hub) to compute NLL and PPL. The average performance (exact matching accuracy) on 25 tasks from Big Bench Hard is in the table below.
>
> | Method    | Zero-shot | LoRA Hub | DaWin (PPL) | DaWin (NLL) |
> |-----------|-------|-------|-------|-------|
> | Avg. perf. | 29.18 | 36.21 | 36.55 | 36.61 |
>
>
> * DaWin using PPL and NLL both show better average performance compared to LoRA Hub which requires additional training per each downstream task. In this setup, compared with the static averaging or zero-shot inference method, DaWin **only requires 13% more memory storage** thanks to the lightweight nature of LoRA.
>
>
> > Q2) Definition of $H$ and a bit of elaboration. Number of model evaluations required for DaWin.
>
> `Answer:`
> * As noted in the caption of Table 2 of the manuscript, $H$ denotes the **size of hyperparameter grid** for the methods that require hyperparameter tuning such as WiSE-FT. A common choices of the hyperparameter sweep range are $\{0.1, 0.15, ..., 0.9, 0.95\}$ or $\{0.1, 0.2, ..., 0.9\}$. Then, $H$ of former and latter cases are 19 and 9, respectively.
> * As the Reviewer sGAj noted, DaWin in robust fine-tuning setup requires **three forward evaluations** given $x$: 1) pre-trained model, 2) fine-tuned model, and 3) post-interpolation model, respectively.
> * Although this is three times larger than a static interpolation method, practically, WiSE-FT requires hyperparameter tuning on the given labeled dataset to achieve competitive performances as noted in the Tables of the manuscript. Therefore, the actual runtime of DaWin would not severely be larger than WiSE-FT as analyzed in Figure 8 of the manuscript.

---

> > ### Comment · Reviewer_sGAj · 2024-11-26
> > **Thank you**
> >
> > Thank you for your explanations and additional experiments. I raised my score.

---

> > > ### Author Response · Authors · 2024-11-26
> > >
> > > Dear Reviewer sGAj,
> > >
> > > We appreciate your re-rating and are sincerely thankful for you taking your valuable time to review, and giving us constructive comments that have remarkably improved the quality of the manuscript!
> > >
> > > Best,
> > >
> > > The authors

---

> ### Author Response · Authors · 2024-11-22
> **Response to the Reviewer sGAj [3/3]**
>
> > Q3) In the experiments, DaWin sacrifices ID performance for better OOD performance compared to WiSE-FT. Could you elaborate in which application scenario I would want to choose DaWin over WiSE-FT. I would think in practice good ID performance still wins over good OOD performance if you cannot have both.
>
> `Answer:` As the Reviewer sGAj noted, in robust fine-tuning experiments, DaWin slightly underperforms WiSE-FT in terms of some ID performances although remarkably outperforms on OOD datasets.
> * This is because when we use the WiSE-FT method, its interpolation coefficient is tuned on the ID validation set to maximize the ID validation accuracy by assuming we can access some labeled ID samples.
> * Meanwhile, DaWin does not assume accessibility on labeled samples by default and determines the interpolation coefficients based on individual models' entropy which can be not always well-correlated with the true model's expertise on a target dataset.
> * We would recommend using DaWin rather than WiSE-FT (or other static merging methods) in the following scenarios.
>   1. A setup where we **can not access any labeled data samples from ID** that can be leveraged as a prior for model design.
>   2. A setup where the model is **expected to encounter OOD data only during the evaluation** phase.
>
> We provide the empirical results (classification accuracy) for the second scenario in the table below.
>
> | Train dataset    | Eval dataset | WiSE-FT | DaWin  |
> |------------------|--------------|---------|--------|
> | SVHN, GTSRB      | MNIST        | 77.31   | 83.67  |
> | SVHN, EuroSAT    | MNIST        | 77.13   | 84.37  |
> | SVHN, SUN397     | MNIST        | 77.51   | 84.04  |
> | SVHN, DTD        | MNIST        | 78.84   | 83.96  |
> | MNIST, RESISC45  | EuroSAT      | 57.37   | 59.70  |
> | SUN397, RESISC45 | EuroSAT      | 60.33   | 61.00  |
> | SVHN, RESISC45   | EuroSAT      | 56.30   | 59.78  |
> | DTD, RESISC45    | EuroSAT      | 55.30   | 58.41  |
>
> * In this setup, we encounter the samples from unseen (OOD) data distribution during evaluation.
> * We observed that DaWin consistently outperforms WiSE-FT (multiple model extended version) in terms of unseen task generalization accuracy.

---

### Author Response · Authors · 2024-11-22
**Global response**

The authors appreciate all the reviewers (`sGAJ`, `Rb5n`, `uT7T`, `yGMX`) for taking the time to intensively review the manuscript and provide valuable comments that significantly contribute to improving the quality of this paper. This global response contains a summary of strengths highlighted by reviewers, revisions of the manuscript, and the reference for literature that we cited in our individual responses. Individual responses per reviewer address weaknesses and questions, separately.



## Strengths of our work, as highlighted by reviews:
1. [`sGAJ`,`uT7T`] **Clear and understandable presentation of paper**, and intuitive design of method of which effectiveness is supported by well-designed experiments from the pilot study on 'GT' to benchmark experiments.
2. [`Rb5n`,`yGMX`] **Training-free characteristic** of DaWin is beneficial in practice
3. [`sGAJ`,`Rb5n`,`yGMX`] **Efficiency consideration through Beta Mixture Modeling** address inference time overhead (computational complexity) of dynamic interpolation
4. [`Rb5n`] DaWin is **performant both on robust fine-tuning and multi-task learning setups**
5. [`Rb5n`] DaWin does **not require extensive hyperparameter tuning**
6. [`uT7T`,`yGMX`] **A range of promising results** backing up the claims, and application scenarios on dynamic classifier selection and output ensemble are further provided.
7. [`uT7T`] **Trade-off analysis** between the accuracy and runtime
8. [`yGMX`] **Novel use of entropy** helps improve sample-wise performance across diverse tasks.


## Revisions of the manuscript
1. **Page 3, Line 160-161**: We added a clarifying sentence for the use of the term '_oracle_' to distinct it from the expression '_optimal_' to address a concern from reviewer `sGAJ`.
2. **Page 5, Line 254-256**: We additionally highlighted the introducing context and use of temperature scaling for DaWin to address a concern from reviewer `Rb5n`.
3. **Page 5, Line 266**: We put an index for a newly added figure in the Appendix that supports our sample-wise entropy valley hypothesis to address a concern from reviewer `uT7T`.
4. **Page 25 (entire)**: We provide additional empirical results that address concerns from `Rb5n` and `uT7T` which visualize the distance between datasets and entropy valley hypotheses.
5. **Page 26 (entire)**: We added two tables addressing concerns of reviewer `sGAJ` and `Rb5n` that show optimality validation of $\lambda^{*}(x)$ and alternative design choice for per-sample interpolation coefficients.


## Global reference
1. Prefix-Tuning: Optimizing Continuous Prompts for Generation, Li et al. 2021
2. The Power of Scale for Parameter-Efficient Prompt Tuning, Lester et al. 2021
3. LoRA: Low-Rank Adaptation of Large Language Models, Hu et al. 2021
4. LoraHub: Efficient Cross-Task Generalization via Dynamic LoRA Composition, Huang et al. 2023
5. Challenging BIG-Bench Tasks and Whether Chain-of-Thought Can Solve Them, Suzgun et al. 2022
6. Robust fine-tuning of zero-shot models, Wortsman et al. 2021
7. Fine-Tuning can Distort Pretrained Features and Underperform Out-of-Distribution, Kumar et al. 2022
8. Finetune like you pretrain: Improved finetuning of zero-shot vision models, Goyal et al. 2023
9. Florence: A New Foundation Model for Computer Vision, Yuan et al. 2021
10. Socratic Models: Composing Zero-Shot Multimodal Reasoning with Language, Zeng et al. 2022
11. EVA-CLIP: Improved Training Techniques for CLIP at Scale, Sun et al. 2023
12. Finetuned Language Models Are Zero-Shot Learners, Wei et al. 2021
13. CoCa: Contrastive Captioners are Image-Text Foundation Models, Yu et al 2022
14. Visual Instruction Tuning, Liu et al. 2023
15. On Calibration of Modern Neural Networks, Guo et al. 2017
16. Correlation Coefficients: Appropriate Use and Interpretation, Schober et al. 2018
17. Linear Mode Connectivity and the Lottery Ticket Hypothesis, Frankle et al. 2020
18. Model Stock: All we need is just a few fine-tuned models, Jang et al. 2024
19. Language Models are Few-Shot Learners, Brown et al. 2020
20. Explainable, trustworthy, and ethical machine learning for healthcare: A survey, Rasheed et al. 2022

---

### Comment · Area_Chair_uzbR · 2024-11-22
**Interactive Discussions**

Dear Reviewers,

Thank you for your efforts in reviewing this paper. We highly encourage you to participate in interactive discussions with the authors before November 26, fostering a more dynamic exchange of ideas rather than a one-sided rebuttal.

Please feel free to share your thoughts and engage with the authors at your earliest convenience.

Thank you for your collaboration.

Best regards,
ICLR 2025 Area Chair

---

### Meta-Review · Area_Chair_uzbR · 2024-12-20

**Metareview:**

This submission introduces DaWin, a dynamic weight interpolation method that combines pre-trained and fine-tuned models based on per-sample entropy without requiring additional training. It further utilizes Beta mixture models to efficiently model interpolation coefficients, ensuring computational efficiency. DaWin demonstrates significant performance improvements and offers a scalable solution for adaptable model merging. After the rebuttal, all four reviewers recommended acceptance, and the area chair concurs with their recommendation to accept this submission.

**Additional Comments On Reviewer Discussion:**

Nearly all concerns were addressed during the rebuttal phase, and the authors are encouraged to incorporate these clarifications into the final version.

---

### Decision · Program_Chairs · 2025-01-22

Accept (Poster)